# Recurrent Submodular Welfare and Matroid Blocking Semi-Bandits

**Orestis Papadigenopoulos**
Department of Computer Science
The University of Texas at Austin
papadig@cs.utexas.edu

**Constantine Caramanis**
Electrical and Computer Engineering
The University of Texas at Austin
constantine@utexas.edu

## Abstract

A recent line of research focuses on the study of stochastic multi-armed bandits (MAB), in the case where temporal correlations of specific structure are imposed between the player's actions and the reward distributions of the arms. These correlations lead to (sub-)optimal solutions that exhibit interesting dynamical patterns – a phenomenon that yields new challenges both from an algorithmic as well as a learning perspective. In this work, we extend the above direction to a combinatorial semi-bandit setting and study a variant of stochastic MAB, where arms are subject to matroid constraints and each arm becomes unavailable (blocked) for a fixed number of rounds after each play. A natural common generalization of the state-of-the-art for blocking bandits, and that for matroid bandits, only guarantees a $1/2$-approximation for general matroids. In this paper we develop the novel technique of correlated (interleaved) scheduling, which allows us to obtain a polynomial-time $(1 - 1/e)$-approximation algorithm (asymptotically and in expectation) for any matroid. Along the way, we discover an interesting connection to a variant of Submodular Welfare Maximization, for which we provide (asymptotically) matching upper and lower approximability bounds. In the case where the mean arm rewards are unknown, our technique naturally decouples the scheduling from the learning problem, and thus allows to control the $(1 - 1/e)$-approximate regret of a UCB-based adaptation of our online algorithm.

## 1 Introduction

Despite the large number of variants of the stochastic *multi-armed bandits* (MAB) model [46, 33] that have been introduced [8, 34], the majority of the results comply with the common assumption that playing an action does not alter the environment, namely, the reward distributions of the subsequent rounds (with notable exceptions discussed below). Only recently, researchers have focused their attention on settings where temporal dependencies of specific structure are imposed between the player's actions and the reward distributions [27, 10, 7, 39, 6]. In [27], Kleinberg and Immorlica consider the setting of *recharging bandits*, where the expected reward of each arm is a concave and weakly increasing function of the time passed since its last play, modeling in that way scenarios of local performance loss. In a similar spirit, Basu et al. [7] consider the problem of *blocking bandits*, in which case once an arm is played at some round, it cannot be played again (i.e., it becomes blocked) for a fixed number of consecutive rounds. Notice that all the aforementioned examples are variations of the stochastic MAB setting, where the decision maker plays (at most) one arm per time step.

When combinatorial constraints and time dynamics come together, the result is a much richer and more challenging setting, precisely because their interplay creates a complex dynamical structure. Indeed, in the standard combinatorial bandits setting [11], the optimal solution in hindsight is to consistently play the feasible subset of arms of maximum expected reward. However, in the presence

of local temporal constraints on the arms, an optimal (or even suboptimal) solution cannot be trivially characterized– a fact that significantly complicates the analysis, both from the algorithmic as well as from the learning perspective. In this work, we study the following bandit setting– a common generalization of *matroid bandits*, introduced by Kveton et al. [30], and blocking bandits [7]:

**Problem 1.1** (Matroid Blocking Semi-Bandits (MBB)). *We consider a set $\mathcal{A}$ of $k$ arms, a matroid $\mathcal{M} = (\mathcal{A}, \mathcal{I})$, and an unknown time horizon of $T$ rounds. Each arm $i \in \mathcal{A}$ is associated with an unknown bounded reward distribution of mean $\mu_i$, and with a known deterministic delay $d_i$, such that whenever an action $i$ is played at some round, it cannot be played again for the next $d_i - 1$ rounds. At each time step, the player pulls a subset of the available (i.e., not blocked) arms restricted to be an independent set of $\mathcal{M}$. Subsequently, she observes the reward realization of each arm played (semi-bandit feedback) and collects their sum as the reward for this round. The goal of the player is to maximize her expected cumulative reward over $T$ rounds.*

The above model captures a number of applications, varying from team formation to ad placement, when arms represent actions that cannot be played repeatedly without restriction. As a concrete example, consider a recommendation system that repeatedly suggests a variety of products (e.g., songs, movies, books) to a user. The need for diversity on the collection of suggested products (arms), to capture different aspects of user's preferences, can be modeled as a linear matroid. Further, the blocking constraints preclude the incessant recommendation of the same product (which can be detrimental, as the product might be perceived as a "spam"), while the maximum rate of recommendation (controlled by the delay) might depend on factors such as popularity, promotion and more. Finally, the expected reward of each product is the probability of purchasing (or clicking).

From a technical viewpoint, the MBB problem is already NP-hard for the simple case of a uniform rank-1 matroid (see Theorem 2.1 in [43]), even in the *full-information* setting, where the reward distributions are known to the player a priori. The natural common generalization of the algorithms in [7, 30], computes and plays, at each time step, an independent set of maximum mean reward consisting of the available elements. While the above strategy is a $(1 - 1/e)$-approximation asymptotically (that is, for $T \to \infty$) for partition matroids, unfortunately, it only guarantees a $1/2$-approximation for general matroids [1] and this guarantee is tight (see Appendix E for an example). A natural question that arises is whether a $(1 - 1/e)$-approximation is possible for any matroid.

The main result of this paper shows that this is indeed possible. Along the way, we identify that the key insight (and also the weak point of the naive $1/2$-approximation) is the underlying *diminishing returns* property hidden in the matroid structure. In particular, we discover an interesting connection of MBB to the following problem of interest in its own right:

**Problem 1.2** (Recurrent Submodular Welfare (RSW)). *We consider a monotone (non-decreasing) submodular function $f : 2^{\mathcal{A}} \to \mathbb{R}_{\geq 0}$ over a universe $\mathcal{A}$ and a time horizon $T$. At each round $t \in [T]$ we choose a subset $\mathcal{A}_t \subseteq \mathcal{A}$ and collect a reward $f(\mathcal{A}_t)$. However, using an element $i \in \mathcal{A}$ at some round $t \in [T]$ makes it unavailable (i.e., blocked) for a fixed and known number of $d_i - 1$ subsequent rounds, namely, during the interval $[t, t + d_i - 1]$. The objective is to maximize $\sum_{t \in [T]} f(\mathcal{A}_t)$, subject to the blocking constraints, within a (potentially unknown) time horizon $T$.*

For the above model, which can be thought of as a variant of *Submodular Welfare Maximization* [47], we provide an efficient randomized $(1 - 1/e)$-approximation (asymptotically), accompanied by a matching hardness result. Note that the RSW problem is a very natural model, capturing applications of submodular maximization in repeating scenarios, where the elements cannot be constantly used without restriction. As an example, consider the process of renting goods to a stream of customers with identical submodular utility functions modeling their satisfaction.

As we show, our approach for the RSW problem immediately implies an algorithm of the same approximation guarantee for the full-information case of MBB and, additionally, it has important implications for the *bandit* setting, where the reward distributions are initially unknown. The standard goal in this case is to provide a (sublinear in the time horizon) upper bound on the *regret*, namely, the difference between the expected reward of a bandit algorithm and a (near-)optimal algorithm, due to the initial lack of knowledge of the former[1].

---

[1] In fact, we upper bound the $(1 - 1/e)$-(approximate) regret, defined as the difference between $(1 - 1/e) \, \text{OPT}(T)$ and the expected reward collected by a bandit algorithm. The notion of $\alpha$-regret is widely used in the combinatorial bandits literature [15, 48] for combinatorial problems where an efficient algorithm

## 1.1 Related Work

A recent line of research focuses on non-stationary models in the case where each reward distribution is a special function of the player's actions [10, 39, 6]. In [7], Basu et al. provide a greedy $(1 - 1/e)$-approximation for the full-information case of the blocking bandits problem (a special case of the MBB model for a uniform rank-1 matroid). As we have already mentioned, generalizing their strategy to the MBB problem fails to provide the same guarantee for general matroids. In the bandit setting, where the reward distributions are initially unknown, the authors have to overcome the burden of characterizing a (sub)optimal solution, where the rate of mean collected reward exhibits significant fluctuations over time. The key insight is to observe that every time the full-information algorithm plays an arm, its bandit variant, which relies on estimations of the mean rewards, has at least one chance of playing the same arm. However, this key coupling argument, that enables sublinear regret bounds, becomes significantly more involved in the presence of matroid constraints.

In [27], Kleinberg and Immorlica study the case of recharging bandits. Their approach first computes the "optimal" playing frequency $1/x_i$ of each arm $i$ via a mathematical formulation. In order to play each arm with this frequency, they develop the technique of *interleaved rounding*, where they associate each arm $i$ with a sequence of real numbers $\{(\alpha_i + k)/x_i\}_{k \in \mathbb{N}}$, with $\alpha_i \sim U[0, 1]$. Then, the arms are played sequentially in the same order they appear on the real line. This novel rounding technique exhibits reduced variance and, thus, an improved approximation guarantee comparing to other natural approaches such as independent randomized rounding.

The MBB model is also related to the literature on *periodic scheduling* [5, 4]. In [43], Sgall et al. consider the problem of periodically scheduling jobs on a set of machines. Each job is associated with a *processing time*, during which it occupies the machine it is executed on, a *vacation time*, namely, a minimum time required after its completion in order to be rescheduled, and a *reward*. It is not hard to see that the case of unit processing times is a special case of MBB with a uniform matroid of rank equal to the number of machines, under the objective of maximizing the total reward. Further, it is known [7] that the rank-1 case of MBB generalizes the *Pinwheel Scheduling* problem [24]: Given $k$ colors associated a set of integers $\{d_i\}_{i \in [k]}$, such that $\sum_{i \in [k]} 1/d_i = 1$, decide whether there is a coloring of the natural numbers $\nu : \mathbb{N} \to [k]$ such that every color $i \in [k]$ appears at least once every $d_i$ numbers. As it is proved in [25], the above problem does not admit a pseudopolynomial time algorithm unless SAT can be solved by a randomized algorithm in expected quasi-polynomial time.

In a concurrent work [1], the authors consider the blocking bandit model in a generic combinatorial setting and under stochastic delays. As they show, the greedy algorithm that plays at each time the maximum feasible subset of available arms is a $\mathcal{O}(1)$-approximation for downward-closed set systems. However, when specialized to matroids, they cannot do better than a $1/2$-approximation. We need new ideas to reach a $(1 - 1/e)$-approximation algorithm and associated regret guarantees for the rich class of matroid bandits.

Our work is related to the line of research regarding bandit optimization of submodular functions (see [13, 20] and references therein). We refer the reader to Appendix A for additional related work on *non-stationary* bandits, *combinatorial* bandits, and *submodular welfare maximization*.

## 1.2 Our Contributions

**Reducing full-information MBB to RSW.** We first focus on the full-information variant of MBB, where the mean rewards of the arms are known to the player a priori. We assume that the player has access to the matroid $\mathcal{M}$ via an independence oracle and knowledge of the arms' fixed delays, yet she is oblivious to the time horizon $T$. In this sense, she plays *online*. An interesting aspect of dynamics, as illustrated in [27, 7, 6], is that one needs to guarantee, via *scheduling*, that each arm is roughly played at a frequency close to its "optimal" rate. This is particularly important in the presence of "hard" blocking constraints, where no reward can be obtained by a blocked arm.

In order to address the above scheduling problem, we propose a particular "decoupled" *two-phase strategy*. We refer to each phase as (cooperative) Player A and Player B. Initially, Player A decides on a schedule that determines arm availability, namely, a subset of rounds where each arm is allowed to be played. Subsequently, Player B chooses a subset of available arms that maximizes the total

---

does not exist, and, thus, any efficient algorithm would inevitably suffer linear regret in standard definition (where $\alpha = 1$).

expected reward, subject to the matroid constraints. In order to completely decouple the two phases, the availability schedule produced by Player A is never affected by which arms are eventually chosen by Player B (that is, it is impossible for Player B to violate the blocking constraints).

In the case where Player B knows the expected rewards of the arms and due to the above decoupling property, his optimal strategy (given any availability schedule) can be easily characterized: Since the arms of each round are subject to matroid constraints, Player B achieves his goal by playing a maximum expected reward independent set among the available arms of each round, which can be computed efficiently using the greedy algorithm for matroids. Thus, the role of Player A becomes to choose an availability schedule that maximizes the total reward, knowing that Player B will behave exactly as described above. The key observation is that the solution computed by Player B at each round, corresponds to the *weighted rank function* of the matroid evaluated on the set of available arms of the round. More importantly, it can be proved that this function is monotone submodular and, hence, Player A's task is a special case of the RSW problem.

**Optimal approximation for RSW.** Focusing our attention on the RSW problem, any "good" solution should guarantee that each element $i \in \mathcal{A}$ is selected a fraction of the time close to $1/d_i$ (the maximum possible), where $d_i$ is the delay. However, a naive randomized approach that selects (if available) each element $i$ with probability $1/d_i$ independently at each round, can be as bad as a $(1 - e^{-1/2}) \approx 0.393$-approximation (see Appendix E for an example). Instead, motivated by the rounding technique of Kleinberg and Immorlica [27], we develop a (time-)correlated sampling strategy, which we call *interleaved scheduling*. While our technique is based on the same principle of transforming (randomly interleaved) sequences of real numbers into a feasible schedule, our implementation is very different. Indeed, as opposed to [27], we additionally face the issue of scheduling more than one arms per round, subject to matroid constraints, and the fact that our "hard" blocking constraints are particularly sensitive to the variance of the produced schedule. Using our technique, we construct a polynomial-time randomized algorithm, named INTERLEAVED-SUBMODULAR (IS), that achieves the following guarantee for RSW:

**Theorem 1.3.** *The expected reward collected by* INTERLEAVED-SUBMODULAR *over $T$ rounds, $\mathcal{R}^{IS}(T)$, is at least $(1 - 1/e) \operatorname{OPT}(T) - \mathcal{O}(d_{\max} f(\mathcal{A}))$, where $\operatorname{OPT}(T)$ is the optimal reward of RSW for $T$ rounds and $d_{\max} = \max_{i \in \mathcal{A}} d_i$ is the maximum delay of the instance.*

The proof of the above guarantee relies on the construction of a *convex program* (CP), based on the *concave closure* of $f$ (see below), that yields an (approximate up to an additive term) upper bound on the optimal reward. Although our algorithm never computes an optimal solution to this convex program, it allows us to compare its expected collected reward with the optimal solution of CP, leveraging known results on the *correlation gap* of submodular functions. As we show via a reduction from the SWM problem with identical utilities, the $(1 - 1/e)$ term in the above guarantee is asymptotically the best possible, unless P = NP; further, the additive term results from the fact that our algorithm is oblivious to the time horizon $T$.

**Bandit algorithm and regret guarantees.** We now turn our attention to the *bandit setting* of MBB, where the mean rewards are initially unknown. Our interleaved scheduling method exhibits an additional property: *It does not rely on the monotone submodular function itself*, a fact that is particularly important for the bandit setting. Indeed, in the full-information setting Player B computes a maximum expected reward independent set at each round, for any availability schedule provided by Player A. In the bandit setting, however, the reward distributions are not a priori known and, thus, must be learned. Nevertheless, we do not need to wait to learn these distributions to find a good availability schedule. This allows us to make a natural coupling between the strategy of Player B in the bandit and in the full-information case and, thus, to compare the expected reward collected "pointwise", assuming a fixed common availability schedule. We remark that the above coupling is very different than the one in [7], as ours is independent of the trajectory of the observed rewards.

The above analysis allows us to develop a bandit algorithm for MBB based on the UCB method, called INTERLEAVED-UCB (IB). Specifically, given any availability schedule provided by Player A (independently of the rewards) and in increasing order of rounds, Player B greedily computes a maximal independent set consisting the available arms of each round, based on estimates (known as UCB indices) of the mean rewards. In order to analyze the regret, we use the independence of the availability schedule in combination with the *strong basis exchange* property of matroids. This allows us to decompose the overall regret of our algorithm into contributions from each individual arm.

Once we have established this regret decomposition, we can bound the individual regret attributed to each arm using more standard UCB type arguments [30], leading to the following guarantee:

**Theorem 1.4.** *The expected reward collected by* INTERLEAVED-UCB *in $T$ rounds, $\mathcal{R}^{IB}(T)$, for $k$ arms, a matroid of rank $r = \mathrm{rk}(\mathcal{M})$ and maximum delay $d_{\max}$ is at least*

$$\left(1 - \frac{1}{e}\right) \mathrm{OPT}(T) - \mathcal{O}\left(k\sqrt{T \ln(T)} + k^2 + d_{\max}r\right).$$

In the above bound, the additive loss corresponds to the regret with respect to $\left(1 - \frac{1}{e}\right)\mathrm{OPT}(T)$. Interestingly, our regret bound is very close (even in constant factors) to the information-theoretically optimal bound provided in [30] for the non-blocking setting. In fact, except for the small additive $\mathcal{O}(d_{\max}r)$ term, the regret bound in [30] is the same as ours, if we replace the number of arms $k$ with $\sqrt{k \cdot r}$. Intuitively, this is due to the fact that our algorithm must learn the complete order of mean rewards, as opposed to the non-blocking setting where learning the maximum expected reward independent set in hindsight is sufficient for eliminating the regret.

All the omitted proofs of our results have been moved to the Appendix.

## 2 Preliminaries on Matroids and Submodular Functions

**Continuous extensions and correlation gap of submodular functions.** Consider any set function $f : 2^{\mathcal{A}} \to \mathbb{R}_{\geq 0}$ over a ground set $\mathcal{A}$. Recall that $f$ is submodular, if $\forall S, T \subseteq \mathcal{A}$ we have $f(S \cup T) + f(S \cap T) \leq f(S) + f(T)$. For any point $\mathbf{x} \in [0,1]^k$, we denote by $S \sim \mathbf{x}$ the random set $S \subseteq \mathcal{A}$, such that $\mathbb{P}(i \in S) = x_i$. We consider two canonical continuous extensions of a set function:

**Definition 2.1** (Continuous extensions). *For any set function $f$ the* multi-linear extension *is*

$$F(\mathbf{x}) = \underset{S \sim \mathbf{x}}{\mathbb{E}}[f(S)] = \sum_{S \subseteq \mathcal{A}} f(S) \prod_{i \in S} x_i \prod_{i \notin S}(1 - x_i).$$

*Moreover, the* concave closure *is defined as*

$$f^+(\mathbf{x}) = \max_{\alpha}\{\sum_{S \subseteq \mathcal{A}} \alpha_S f(S) \mid \sum_{S \subseteq \mathcal{A}} \alpha_S \mathbf{1}_S = \mathbf{x}, \sum_{S \subseteq \mathcal{A}} \alpha_S = 1, \alpha \succeq 0\},$$

*where $\mathbf{1}_S \in \{0,1\}^k$ is an indicator vector such that $(\mathbf{1}_S)_i = 1$, if $i \in S$, and $(\mathbf{1}_S)_i = 0$, otherwise.*

**Lemma 2.2** (Correlation gap [9]). *Let $f : 2^k \to \mathbb{R}_{\geq 0}$ be a monotone (non-decreasing) submodular function. Then for any point $\mathbf{x} \in [0,1]^k$, we have*

$$F(\mathbf{x}) \leq f^+(\mathbf{x}) \leq (1 - 1/e)^{-1} F(\mathbf{x}).$$

**Matroids and weighted rank functions.** Consider a matroid $\mathcal{M} = (\mathcal{A}, \mathcal{I})$, where $\mathcal{A}$ is the *ground set* and $\mathcal{I}$ is the family of *independent sets*. Recall that in any matroid, the family $\mathcal{I}$ satisfies the following two properties: (i) Every subset of an independent set (including the empty set) is an independent set, namely, if $S' \subset S \subseteq \mathcal{A}$ and $S \in \mathcal{I}$, then $S' \in \mathcal{I}$ (*hereditary property*). (ii) Let $S, S' \subseteq \mathcal{A}$ be two independent sets with $|S| < |S'|$, then there exists some $e \in S' \backslash S$ such that $S \cup \{e\} \in \mathcal{I}$ (*augmentation property*). See [42, 37] for more details on matroids.

We assume that access to $\mathcal{M}$ is given through an *independence oracle* [22, 40], namely, a black-box routine that, given a set $S \subseteq \mathcal{A}$, answers whether $S$ is an independent set of $\mathcal{M}$. For any set $R \subset \mathcal{A}$ we define the *restriction* of $\mathcal{M}$ to $R$, denoted by $\mathcal{M}|R$, to be the matroid $\mathcal{M}|R = (R, \{I \in \mathcal{I} \mid I \subseteq R\})$.

Given any non-negative linear *weight* vector $\mathbf{w} \in \mathbb{R}_{\geq 0}^k$, the problem of computing a maximum weight independent set can be solved optimally by the standard greedy algorithm: Starting from the empty set $S = \emptyset$, add each ground element $e \in \mathcal{A}$ to the set $S$ in a non-increasing order of weights, as long as the set $S \cup \{e\}$ does not contain a circuit. Given a matroid $\mathcal{M} = (\mathcal{A}, \mathcal{I})$ and a weight vector $\mathbf{w}$, the function $f_{\mathcal{M}, \mathbf{w}}(S) = \max_{I \in \mathcal{I}, I \subseteq S}\{\mathbf{w}(I)\}$ is called the *weighted rank function* of $\mathcal{M}$ and returns the weight of the maximum independent set of the restriction $\mathcal{M}|S$.

**Lemma 2.3** (Weighted rank function [9]). *For any matroid $\mathcal{M}$ and non-negative weight vector $\mathbf{w}$, the function $f_{\mathcal{M}, \mathbf{w}}(S) = \max_{I \in \mathcal{I}, I \subseteq S}\{\mathbf{w}(I)\}$ is monotone (non-decreasing) submodular.*

**Technical notation.** For any event $\mathcal{E}$, we denote by $\mathcal{X}(\mathcal{E}) \in \{0,1\}$ the indicator variable such that $\mathcal{X}(\mathcal{E}) = 1$, if $\mathcal{E}$ occurs, and $\mathcal{X}(\mathcal{E}) = 0$, otherwise. For any non-negative integer $n \in \mathbb{N}$, we define $[n] = \{1, 2, \ldots, n\}$. For any vector $\mu \in \mathbb{R}^k$ and set $S \subseteq [k]$, we define $\mu(S) = \sum_{i \in S} \mu_i$. Moreover, we use the notation $t \in [a, b]$ (for $a \leq b$) for some time index $t$, in place of $t \in [T] \cap [a, \ldots, b]$. Unless otherwise noted, we use the indices $i$, $j$ or $i'$ to refer to arms and $t$, $t'$ or $\tau$ to refer to time. Let $\mathcal{A}_t^\pi \in \mathcal{I}$ be the set of arms played by some algorithm $\pi \in \{IS, IG, IB\}$ (defined in Sections 3 and 4) at time $t$. Unless otherwise noted, all expectations are taken over the randomness of the offsets $\{r_i\}_{i \in [k]}$ (see Section 3) and the reward realizations.

# 3  Recurrent Submodular Welfare

Let $f(S) : 2^\mathcal{A} \to \mathbb{R}_{\geq 0}$ be a monotone submodular function over a universe $\mathcal{A}$ of $k$ elements, such that $f(\emptyset) = 0$. In the *blocking* setting, each element $i \in \mathcal{A}$ is associated with a known deterministic *delay* $d_i \in \mathbb{N}_{>0}$, such that once the arm is played at some round $t$, it becomes unavailable for the next $d_i - 1$ rounds, namely, in the interval $\{t, \ldots, t + d_i - 1\}$. At each round $t \in [T]$, the player chooses a subset $\mathcal{A}_t$ of available (i.e., non-blocked) elements and collects a reward $f(\mathcal{A}_t)$. The goal is to maximize the total reward collected, i.e., $\sum_{t \in [T]} f(\mathcal{A}_t)$, within an unknown time horizon $T$.

We provide an efficient randomized $(1 - 1/e)$-approximation algorithm for RSW. Informally, the algorithm starts by considering, for each element $i \in \mathcal{A}$, a sequence of rational numbers of the form $\{t \cdot \frac{1}{d_i}\}_{t \in [T]}$. Then, these sequences are *interleaved* by randomly adding an *offset* $r_i$, drawn uniformly at random from $[0,1]$, for each $i \in \mathcal{A}$ to the corresponding sequence. At every round $t \in [T]$, the algorithm chooses a set $\mathcal{A}_t$, consisting only of elements for which the (perturbed) interval $L_{i,t} = [t \cdot \frac{1}{d_i} + r_i, (t+1) \cdot \frac{1}{d_i} + r_i)$ contains an integer.

**Algorithm 3.1** (INTERLEAVED-SUBMODULAR (IS)). *For each element $i \in \mathcal{A}$, let $r_i \sim U[0,1]$ be a random offset drawn uniformly from $[0,1]$. At every round $t = 1, 2, \ldots$, let $\mathcal{A}_t \subseteq \mathcal{A}$ be the subset of elements such that for any $i \in \mathcal{A}_t$, the interval $L_{i,t} = [t \cdot \frac{1}{d_i} + r_i, (t+1) \cdot \frac{1}{d_i} + r_i)$ contains an integer. Choose the elements $\mathcal{A}_t$ and collect the reward $f(\mathcal{A}_t)$.*

## 3.1  Correctness and approximation guarantee.

We first show the algorithm is correct, namely, that the elements chosen at each round respect the blocking constraints. The correctness is established by the following simple observation:

**Fact 3.2.** *At any $t \in [T]$, all the elements in $\mathcal{A}_t$ are available (i.e., not blocked).*

In order to prove the competitive guarantee of our algorithm, we first construct a convex programming (CP)-based (approximate) upper bound on the optimal reward. Although our algorithm never computes an optimal solution to this CP, this step allows us to prove our guarantee, leveraging results on the correlation gap of submodular functions. For $\boldsymbol{d}^{-1} \in \mathbb{R}^k$ such that $(\boldsymbol{d}^{-1})_i = 1/d_i, \forall i \in [k]$, consider the following formulation based on the concave closure $f^+$ of $f$:

$$\underset{\mathbf{z} \in \mathbb{R}^k}{\text{maximize}}: \ T \cdot f^+(\mathbf{z}) \ \text{ s.t. } \ \mathbf{0} \preceq \mathbf{z} \preceq \boldsymbol{d}^{-1}. \tag{CP}$$

In (CP), each variable $z_i$ can be thought of as the fraction of rounds where element $i \in \mathcal{A}$ is chosen. Intuitively, the constraints indicate the fact that, due to the blocking, each element $i \in \mathcal{A}$ can be played at most once every $d_i$ steps. In order to derive (CP), we start from a non-convex integer program (IP) with 0-1 variables $\{x_{i,t}\}_{i \in \mathcal{A}, t \in [T]}$, each indicating whether element $i \in \mathcal{A}$ is used at round $t \in [T]$. The objective is to maximize $\sum_{t \in [T]} \sum_{S \subseteq \mathcal{A}} f(S) \prod_{i \in S} x_{i,t} \prod_{i \notin S} (1 - x_{i,t})$ subject to natural blocking constraints. For integral solutions, the above objective is equivalent to $\sum_{t \in [T]} f^+(\mathbf{x}_t)$ (where $(\mathbf{x}_t)_i = x_{i,t}$) and, thus, the above relaxation is simply the result of averaging over time the variables and constraints of this IP. By using the concavity of $f^+$, we are able to show that (CP) yields an (approximate) upper bound on the optimal solution of RSW, while the approximation becomes exact as $T$ increases.

**Lemma 3.3.** *Let $\mathcal{R}^{CP}(T)$ be the optimal solution to (CP) and $\mathrm{OPT}(T)$ be the optimal solution over $T$ rounds. We have $\mathcal{R}^{CP}(T) \geq \mathrm{OPT}(T) - \mathcal{O}(d_{\max} f(\mathcal{A}))$, where $d_{\max} = \max_{i \in \mathcal{A}}\{d_i\}$.*

Before we complete the proof of our first main result, we first compute the probability that $i \in \mathcal{A}_t$, i.e., an element $i \in \mathcal{A}$ is sampled at round $t \in [T]$:

**Fact 3.4.** *For any $i \in \mathcal{A}$ and $t \in [T]$, we have $\mathbb{P}(i \in \mathcal{A}_t) = \mathbb{P}(L_{i,t} \cap \mathbb{N} \neq \emptyset) = 1/d_i$.*

*Proof of Theorem 1.3.* Let us denote by $S \sim \mathbf{p}$ with $\mathbf{p} \in [0,1]^k$ the random set $S \subseteq \mathcal{A}$, where each element $i$ participates in $S$ independently with probability equal to $p_i$. By Fact 3.4 and due to the randomness of the offsets $\{r_i\}_{i \in \mathcal{A}}$, we have that $\mathcal{A}_t \sim \mathbf{d}^{-1}$ for each $t \in [T]$. Let $\mathbf{z}^*$ be an optimal solution to (CP). By monotonicity of $f$ and the fact that $\mathbf{z}^* \preceq \mathbf{d}^{-1}$, for the expected value of $f(\mathcal{A}_t)$ at any round $t \in [T]$, we know that $\mathop{\mathbb{E}}_{\mathcal{A}_t \sim \mathbf{d}^{-1}} [f(\mathcal{A}_t)] \geq \mathop{\mathbb{E}}_{\mathcal{A}_t \sim \mathbf{z}^*} [f(\mathcal{A}_t)]$. Moreover, by definition of the multi-linear extension, we have that $\mathop{\mathbb{E}}_{\mathcal{A}_t \sim \mathbf{z}^*} [f(\mathcal{A}_t)] = F(\mathbf{z}^*)$, while by Lemma 2.2 (the correlation gap of submodular functions), we have that, $F(\mathbf{z}) \geq \left(1 - \frac{1}{e}\right) f^+(\mathbf{z})$ for any vector $\mathbf{z} \in [0,1]^k$. By combining the above facts, we can see that

$$\mathcal{R}^{IS}(T) = \sum_{t \in [T]} \mathop{\mathbb{E}}_{\mathcal{A}_t \sim \mathbf{d}^{-1}} [f(\mathcal{A}_t)] \geq \sum_{t \in [T]} F(\mathbf{z}^*) \geq \left(1 - \frac{1}{e}\right) T \cdot f^+(\mathbf{z}^*) = \left(1 - \frac{1}{e}\right) \mathcal{R}^{CP}(T).$$

Therefore, by Lemma 3.3, we can conclude that $\mathcal{R}^{IS}(T) \geq \left(1 - \frac{1}{e}\right) \mathrm{OPT}(T) - \mathcal{O}(d_{\max} f(\mathcal{A}))$. $\quad\square$

In Appendix C.2, we provide a $(1 - 1/e)$-hardness result for RSW, thus proving that the guarantee of Theorem 1.3 is asymptotically tight. This result, which holds even for the special case where $d_{\max} = o(T)$ (that is when the delays are significantly smaller than the time horizon), is proved via a reduction from the SWM problem with identical utilities, in a way that the constructed RSW instance accepts w.l.o.g. solutions of a simple periodic structure.

**Theorem 3.5.** *For any $\epsilon > 0$, there exists no polynomial-time $\left(1 - \frac{1}{e} + \epsilon\right)$-approximation algorithm for the RSW problem, unless $\mathbf{P} = \mathbf{NP}$, even in the special case where $d_{\max} = o(T)$.*

## 4 Matroid Blocking Semi-Bandits

Let $\mathcal{A}$ be a set of $k$ arms and $T$ be an unknown time horizon. At any round $t \in [T]$ and for each $i \in \mathcal{A}$ a reward $X_{i,t}$ is drawn independently from an unknown distribution of mean $\mu_i$ and bounded support in $[0,1]$. Let $d_i \in \mathbb{N}_{>0}$ be the known determinisitc delay of each arm $i \in \mathcal{A}$, and $d_{\max} = \max_{i \in \mathcal{A}} \{d_i\}$. At any round $t \in [T]$, the player pulls any subset $\mathcal{A}_t$ of the available (i.e., non-blocked) arms, as long as it forms an independent set of a given matroid $\mathcal{M} = (\mathcal{A}, \mathcal{I})$. The player only observes the realized reward of each arm she plays and collects their sum. The goal is to maximize the *expected cumulative reward* collected within $T$ rounds, denoted by $\mathcal{R}^{IG}(T) = \mathbb{E}\left[\sum_{t \in [T]} \sum_{i \in \mathcal{A}} X_{i,t} \mathcal{X}(i \in \mathcal{A}_t)\right]$.

**The full-information setting** The following algorithm is the implementation of IS in the special case of the full-information MBB setting, where the mean rewards $\{\mu_i\}_{i \in \mathcal{A}}$ are known a priori:

**Algorithm 4.1** (INTERLEAVED-GREEDY (IG))**.** *For each arm $i \in \mathcal{A}$, let $r_i \sim U[0,1]$ be a random offset drawn uniformly from $[0,1]$. At every round $t = 1, 2, \ldots$, let $\mathcal{G}_t \subseteq \mathcal{A}$ be the subset of arms $i \in \mathcal{A}$, such that the interval $L_{i,t} = [t \cdot \frac{1}{d_i} + r_i, (t+1) \cdot \frac{1}{d_i} + r_i)$ contains an integer. Greedily compute a maximum independent set $\mathcal{A}_t$ of $\mathcal{M} \mid \mathcal{G}_t$ with respect to $\{\mu_i\}_{i \in \mathcal{G}_t}$ and play these arms.*

The following result is an immediate corollary of Theorem 1.3, given that the value of the greedily computed maximum independent set in $\mathcal{M} \mid \mathcal{G}_t$ corresponds to the weighted rank function $f_{\mathcal{M}, \mu}(\mathcal{G}_t)$ which, by Lemma 2.3, is monotone submodular:

**Theorem 4.2.** *The expected reward collected by INTERLEAVED-GREEDY for $T$ rounds, $\mathcal{R}^{IG}(T)$, is at least $\left(1 - \frac{1}{e}\right) \mathrm{OPT}(T) - \mathcal{O}(d_{\max} \mathrm{rk}(\mathcal{M}))$, where $\mathrm{OPT}(T)$ is the optimal expected reward.*

**Remark 4.3.** *The analysis of $IG$ is tight for rank-1 matroids. Indeed, consider $k$ arms, each of delay $k$ and deterministic reward equal to $1$. For $T \to \infty$, the optimal average reward is equal to $1$, simply by playing the arms in a round-robin manner. However, the probability that at least one arm is sampled at some round $t$ is $\sum_{i=1}^{k} \binom{k}{i} \left(\frac{1}{k}\right)^i \left(1 - \frac{1}{k}\right)^{k-i} = 1 - \left(1 - \frac{1}{k}\right)^k \to 1 - \frac{1}{e}$ as $k \to \infty$.*

**The bandit setting and regret analysis** In the setting where the mean rewards are initially unknown, we develop a UCB-based bandit algorithm, INTERLEAVED-UCB (IB). The algorithm is identical to IG, except for the greedy computation of the maximum independent set over the sampled

arms, which is now performed using estimates. Specifically, the algorithm maintains for every $i \in \mathcal{A}$, $t \in [T]$ the following upper estimate of $\mu_i$:

$$\bar{\mu}_{i,t} = \hat{\mu}_{i,T_i(t)} + c_{i,t} \text{ with } c_{i,t} = \sqrt{\frac{2 \ln(t)}{T_i(t)}},$$

where $T_i(t)$ denotes the number of times arm $i$ has been played at the beginning of round $t$ and $\hat{\mu}_{i,T_i(t)}$ denotes the empirical average of the $T_i(t)$ i.i.d. samples from its reward distribution. The term $c_{i,t}$ is the *confidence length* around $\hat{\mu}_{i,T_i(t)}$ that guarantees $\bar{\mu}_{i,t}$ lies in $[\mu_i, \mu_i + 2c_{i,t}]$ with high probability. Note that all the above quantities are random variables depending on the random offsets and the observed reward realizations.

We are interested in upper bounding the $\alpha$-regret, for $\alpha = 1 - \frac{1}{e}$, namely, the difference between $\alpha \text{OPT}(T)$ and the expected reward collected by IB. Due to the complex time dynamics, characterizing the optimal expected reward as a function of the instance is hard. However, using Theorem 4.2 we can upper bound $\alpha \text{OPT}(T)$ by the expected reward collected by IG, thus giving:

$$\alpha \text{OPT}(T) - \mathcal{R}^{UCB}(T) \leq \mathcal{R}^{IG}(T) - \mathcal{R}^{UCB}(T) + \mathcal{O}(d_{\max} \cdot \text{rk}(\mathcal{M})). \tag{1}$$

By the above inequality, it becomes clear that in order to upper bound the regret, it suffices to bound the difference between the expected reward collected by IG and IB. This difference not only depends on the reward realizations (through the UCB estimates), but also on the trajectory of sampled arms in each algorithm, which is itself a function of the random offsets. However, by construction of our interleaved scheduling scheme, these offsets are sampled at the initialization phase of each algorithm and are identically distributed. Thus, the trajectories of sampled arms in the two algorithms exhibit a coupled evolution. This allows us to analyse the regret "pointwise", under the assumption that the sequences of sampled arms are identical throughout the time horizon. To make this idea precise, let $\mathbf{r}^{\pi} \in [0,1]^k$ be the random offsets used and let $\{\mathcal{G}_t^{\pi}(\mathbf{r}^{\pi})\}_{t \in [T]}$ be the sequence of sampled arms by algorithm $\pi \in \{IG, IB\}$. Using (henceforth) $\mathcal{Q}$ to denote the randomness due to the reward realizations of the arms, the next lemma gives our pointwise regret bound.

**Lemma 4.4.** *Let $\bar{\mu}_t(S) = \sum_{i \in S} \bar{\mu}_{i,t}$ and $\mu(S) = \sum_{i \in S} \mu_i$. We have*

$$\mathcal{R}^{IG}(T) - \mathcal{R}^{IB}(T) = \mathop{\mathbb{E}}_{\mathbf{r} \sim U[0,1]^k, \mathcal{Q}} \left[ \sum_{t \in [T]} \left( \max_{S \subseteq \mathcal{G}_t(\mathbf{r}), S \in \mathcal{I}} \{\mu(S)\} - \mu \left( \arg \max_{S \subseteq \mathcal{G}_t(\mathbf{r}), S \in \mathcal{I}} \{\bar{\mu}_t(S)\} \right) \right) \right].$$

Thus w.l.o.g., we focus on the case where the sequences of sampled arms are identical. Let $\mathcal{E}_{\mathbf{r}}$ denote the event that both algorithms, IG and IB, sample the same offset vector $\mathbf{r}$, namely, $\mathbf{r}^{IG} = \mathbf{r}^{IB} = \mathbf{r}$. Assuming that $\mathcal{E}_{\mathbf{r}}$ holds for some $\mathbf{r} \in [0,1]^k$, let $\{\mathcal{G}_t\}_{t \in [T]} = \{\mathcal{G}_t(\mathbf{r})\}_{t \in [T]}$ be the sequence of sampled arms, common in both algorithms. Clearly, IB accumulates regret only when it plays independent sets of arms that are suboptimal w.r.t. the true means, i.e., when $\mu(\mathcal{A}_t^{IB}) < \mu(\mathcal{A}_t^{IG})$ for some $t \in [T]$. We assume w.l.o.g. that the arms are indexed in decreasing order of mean rewards and that these mean rewards are distinct. We now formally define the gaps related to our analysis:

**Definition 4.5** (Gaps). *For any subset $S \subseteq \mathcal{A}$ and reward vector $\nu \in \mathbb{R}^k$, we define*

$$\Delta_S(\nu) = \max_{I \in \mathcal{I}, I \subseteq S} \{\mu(I)\} - \mu \left( \arg \max_{B \in \mathcal{I}, B \subseteq S} \{\nu(B)\} \right).$$

*Moreover, let $\Delta_{i,j} = \mu_i - \mu_j$ be the standard suboptimality gap between two arms $i, j \in \mathcal{A}$.*

By Lemma 4.4 and assuming that the event $\mathcal{E}_{\mathbf{r}}$ holds for some $\mathbf{r}$, we are interested in bounding the expectation of $\sum_{t \in [T]} \Delta_{\mathcal{G}_t(\mathbf{r})}(\bar{\mu}_t)$ w.r.t. the reward realizations. The next step is to decompose the suboptimality of IB by noticing that both algorithms play, at each round $t \in [T]$, a basis of $\mathcal{M} \mid \mathcal{G}_t$ and thus $|\mathcal{A}_t^{IG}| = |\mathcal{A}_t^{IB}|$. We use the following fundamental property of matroids:

**Theorem 4.6** (Strong Basis Exchange, Corollary 39.12a in [42]). *Let $\mathcal{M} = (\mathcal{A}, \mathcal{I})$ be a matroid and $I_1, I_2 \in \mathcal{I}$ be two independent sets such that $|I_1| = |I_2|$. Then, there exists a bijection $\sigma : I_1 \to I_2$, such that for any $i \in I_1$ the set $I_1 - i + \sigma(i)$ is an independent set of $\mathcal{M}$.*

Let $\sigma_t : \mathcal{A}_t^{IB} \to \mathcal{A}_t^{IG}$ for each $t \in [T]$ be the bijection described in Theorem 4.6 with respect to the sets $\mathcal{A}_t^{IB}$ and $\mathcal{A}_t^{IG}$ and let $\sigma_t^{-1}$ be its inverse mapping. Note that in any bijection $\sigma_t$ and any $i \in \mathcal{A}_t^{IB} \cap \mathcal{A}_t^{IG}$ we can assume w.l.o.g. that $\sigma_t(i) = i$. Notice, further, that under the event $\mathcal{E}_{\mathbf{r}}$, the bijections $\{\sigma_t\}_{t \in [T]}$ are still random variables that depend on the observed realizations.

**Lemma 4.7.** *Under the event $\mathcal{E}_{\mathbf{r}}$ and at any time $t \in [T]$, we have $\Delta_{\mathcal{G}_t}(\bar{\mu}_t) = \sum_{i \in \mathcal{A}_t^{IG}} \Delta_{i, \sigma_t^{-1}(i)}$.*

Conditioned on the fact that both algorithms operate on the same sequence $\{\mathcal{G}_t\}_{t \in [T]}$ of sampled arms, Lemma 4.7 allows us to decompose the suboptimality gap $\Delta_{\mathcal{G}_t}(\bar{\mu}_t)$ of each round $t \in [T]$, into simpler gaps of the form $\Delta_{i,j}$ between any arms $i \in \mathcal{A}_t^{IG}$ and $j \in \mathcal{A}_t^{IB}$ that are perfectly matched according to the bijection $\sigma_t$, namely, $\sigma_t(j) = i$. Assuming that the event $\{\sigma_t(j) = i\}$ directly implies that $i \in A_t^{IG}$ and $j \in A_t^{IB}$, we can further upper bound the regret as

$$\sum_{t \in [T]} \Delta_{\mathcal{G}_t}(\bar{\mu}_t) = \sum_{t \in [T]} \sum_{i \in \mathcal{A}_t^{IG}} \Delta_{i, \sigma_t^{-1}(i)} \leq \sum_{t \in [T]} \sum_{i \in \mathcal{A}_t^{IG}} \sum_{j \in \mathcal{A}, \Delta_{i,j} > 0} \Delta_{i,j} \, \mathcal{X}\left(\sigma_t(j) = i\right).$$

The above inequality allows us to study the regret attributed to each arm independently, using more standard arguments for UCB-based algorithms in combination with Theorem 4.6. Specifically, for every pair of arms $i, j \in \mathcal{A}$ with $i < j$ (thus, $\Delta_{i,j} > 0$), we define a threshold $\ell_{i,j}$ with the following key-property: After IB "exchanges" arm $j$ for arm $i = \sigma_t(j)$ more than $\ell_{i,j}$ times, due to insufficient exploration, then it has collected enough samples to infer that $\mu_j < \mu_i$ with high probability.

**Lemma 4.8.** *Let $\ell_{i,j} = \left\lfloor \frac{8 \ln(T)}{\Delta_{i,j}^2} \right\rfloor$ for any $i < j$. Under event $\mathcal{E}_{\mathbf{r}}$ and for any arm $j > 1$, we have*

$$\sum_{t \in [T]} \sum_{i < j} \Delta_{i,j} \, \mathcal{X}\left(\sigma_t(j) = i, T_j(t) \leq \ell_{i,j}\right) \leq \frac{16}{\Delta_{j-1,j}} \ln(T) \qquad \textit{(Under-sampled regret)} \quad (2)$$

$$\mathbb{E}_{\mathcal{Q}}\left[\sum_{t \in [T]} \sum_{i < j} \Delta_{i,j} \, \mathcal{X}\left(\sigma_t(j) = i, T_j(t) > \ell_{i,j}\right)\right] \leq \frac{\pi^2}{3} \sum_{i=1}^{j-1} \Delta_{i,j} \quad \textit{(Sufficiently sampled regret)} \quad (3)$$

*Proof sketch of Theorem 1.4.* By inequality (1) and Lemma 4.4, in order to bound the regret of IB, it suffices to upper bound the difference between $\mathcal{R}^{IG}(T)$ and $\mathcal{R}^{IB}(T)$, conditioned on the fact that both algorithms use exactly the same offset vector $\mathbf{r}$ and, thus, they operate on the exact same sequence of sampled arms, denoted by $\{\mathcal{G}_t\}_{t \in [T]}$. By construction, IG plays at any round $t \in [T]$ a basis of $\mathcal{M} \,|\, \mathcal{G}_t$ of maximum expected reward, while IB plays a basis of $\mathcal{M} \,|\, \mathcal{G}_t$ that is maximum with respect to the estimates $\{\bar{\mu}_{i,t}\}_{i \in \mathcal{A}}$. By Theorem 4.6, we can consider a perfect matching between exchangeable arms of $\mathcal{A}_t^{IG}$ and $\mathcal{A}_t^{IB}$ and, thus, to decompose the regret into suboptimality gaps between individual arms. Then, using Lemma 4.8, we can upper bound on the expected regret due to the fact that IB erroneously plays arm $j$ instead of arm $i$, when $\Delta_{i,j} > 0$. The above analysis culminates in the following *gap-dependent* regret upper bound:

$$\sum_{j > 1} \frac{16}{\Delta_{j-1,j}} \ln(T) + \frac{\pi^2}{3} \sum_{j > 1} \sum_{i=1}^{j-1} \Delta_{i,j} + \mathcal{O}(d_{\max} \cdot \mathrm{rk}(\mathcal{M})) \quad \text{(gap-dependent regret)}.$$

In order to derive a gap-independent regret bound, we partition the gaps into "small" and "large" and notice that any pair of arms $i, j \in \mathcal{A}$ with $\Delta_{i,j} < \Theta(\sqrt{\frac{\ln(T)}{T}})$ cannot contribute more than $\sqrt{T \ln(T)}$ loss in the regret. $\qquad \square$

## Conclusion and Further Directions

We explore the effect of action-reward dependencies in the combinatorial MAB setting by introducing and studying the MBB problem. After relating the problem to RSW, we provide a $(1 - 1/e)$-approximation for its full-information case, based on the technique of interleaved scheduling. Importantly, our technique is oblivious to the reward distributions of the arms– a fact that allows us to provide regret bounds of optimal dependence in $T$, when these distributions are initially unknown. We believe that this idea could be further applied to different classes of (combinatorial) non-stationary bandits, other than blocking bandits.

Our work leaves behind numerous interesting questions. By exhaustive search over $\mathcal{O}(1)$-periodic schedules, one can construct a PTAS for the (asymptotic) MBB problem, assuming *constant* $\mathrm{rk}(\mathcal{M})$ and $\{d_i\}_{i \in [k]}$. It remains an open question, however, whether the $(1 - 1/e)$-approximation is the

best possible in general. We remark that the hardness of MBB cannot solely rely on an argument similar to Theorem 3.5, since the welfare maximization problem for the class of *gross substitutes*, which includes weighted matroid rank functions, is easy [35]. Finally, it is easy to show that our algorithm gives a $\mathcal{O}(1)$-approximation for the case of stochastic delays. Whether we can recover a $(1 - 1/e)$-approximation in this case is an interesting open question.

## Acknowledgements

The authors would like to thank an anonymous reviewer of a previous version of this work for an unusually thoughtful and helpful review, which aided us in improving the document — in particular, for pointing out the idea of correlated rounding. Further, the authors would like to thank Jannik Matuschke for noticing that the weighted matroid rank function falls into the class of gross substitutes.

## Funding Transparency Statement

This research was partially supported by NSF Grant 2019844. In addition, this research was sponsored by the Army Research Office and was accomplished under Cooperative Agreement Number W911NF-19-2-0333. The views and conclusions contained in this document are those of the authors and should not be interpreted as representing the official policies, either expressed or implied, of the Army Research Office or the U.S. Government. The U.S. Government is authorized to reproduce and distribute reprints for Government purposes notwithstanding any copyright notation herein.

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
