## A  Further Related Work

The MBB model belongs to the family of stochastic *non-stationary* bandits, given that the reward distributions of the arms can change over time. Significant members of this family are *restless bandits* [49, 21], where the reward distribution of each arm changes at each time step, and *rested bandits* [19, 45], where the distribution changes only when the arm is played. For the setting of restless bandits and without further assumptions on the transition functions, it is PSPACE-hard to even approximate the optimal solution [38]. Our model differs from the above cases as we consider a transition function of special form and the transitions can occur both during playing and not playing an arm. In addition, the MBB model falls into the category of Markov Decision Processes (MDPs) with deterministic transitions and stochastic rewards, but requires an exponential (in the size of the arms) state space, which makes this approach inefficient in practice.

A rich body of research on combinatorial bandits [17, 15, 14, 32, 31, 48] focuses on bandit optimization problems over general combinatorial structures. In [30], Kveton et al. consider the problem of stochastic combinatorial bandits where the underlying feasible set is a matroid defined over the ground set of arms. At each round, the player pulls an independent subset of arms and collects their realized rewards, assuming *semi-bandit* feedback (as opposed to the pure exploration *full-feedback* variant studied in [12]). The authors develop a greedy algorithm based on the Upper Confidence Bound (UCB) method [2], while they exploit well-known exchange properties of matroids for achieving optimal regret bounds. Their approach relies on the fact that the optimal solution in hindsight is fixed throughout the time horizon– a fact that is no longer true in the presence of blocking constraints. Additional lines of research that are related to, yet incompatible with, our problem are *bandits with knapsacks* [3, 41] or *with budgets* [16, 44], and *sleeping bandits* [28].

The RSW problem is closely related to the problem of *Submodular Welfare Maximization* (SWM) [47, 36, 26, 18]: Given $k$ items and $m$ players, each associated with a monotone submodular utility function $u_i : 2^{[k]} \to \mathbb{R}_{\geq 0}$, the goal is to partition the elements into $m$ sets $S_1, \ldots, S_m$, one for each player, such that to maximize $\sum_{i \in [m]} u_i(S_i)$. Specifically, RSW can be thought of as a version of the SWM problem, when the items are distributed to a (possibly infinite) stream of players with identical utilities, and each item can be reused after some fixed time period (note that this is different than the online setting in [29]). Interestingly, as noted in [47], the SWM problem with identical utilities is *approximation resistant* in the sense that allocating the items to the players uniformly at random achieves the optimal approximation guarantee of $\left(1 - \frac{1}{e}\right)$ for this setting.

## B  Concentration inequalities

**Theorem B.1** (Hoeffding's Inequality [23]). *Let $X_1, \ldots, X_n$ be independent identically distributed random variables with common support in $[0, 1]$ and mean $\mu$. Let $Y = X_1 + \cdots + X_n$. Then for any $\delta \geq 0$,*

$$\mathbb{P}\left(Y - n\mu \geq \delta\right) \leq e^{-2\delta^2/n} \text{ and } \mathbb{P}\left(Y - n\mu \leq -\delta\right) \leq e^{-2\delta^2/n}.$$

## C  Recurrent Submodular Welfare: Omitted Proofs

### C.1  Correctness and approximation guarantee

**Fact 3.2.** *At any $t \in [T]$, all the elements in $\mathcal{A}_t$ are available (i.e., not blocked).*

*Proof.* Recall that at any round $t \in [T]$, the algorithm only chooses a subset $\mathcal{A}_t$ of the elements. Consider any element $i \in \mathcal{A}$ such that $i \in \mathcal{A}_t$ for some $t \in [T]$. By definition of $\mathcal{A}_t$, the interval $L_{i,t} = [t \cdot \frac{1}{d_i} + r_i, (t+1) \cdot \frac{1}{d_i} + r_i)$ contains an integer. It is not hard to see that, in that case, none of the intervals $L_{i,t'}$ for $t' \in [t - d_i + 1, d_i - 1]$ can contain an integer. Therefore, the last time element $i$ has been chosen must be before $t - d_i$, which implies feasibility with respect to the blocking constraints. □

**Fact 3.4.** *For any $i \in \mathcal{A}$ and $t \in [T]$, we have $\mathbb{P}\left(i \in \mathcal{A}_t\right) = \mathbb{P}\left(L_{i,t} \cap \mathbb{N} \neq \emptyset\right) = 1/d_i.$*

*Proof.* For any fixed $i \in \mathcal{A}$ and $t \in [T]$, because of the fact that $\frac{1}{d_i} \leq 1$ and $r_i \in [0, 1]$, the interval $L_{i,t} = [t \cdot \frac{1}{d_i} + r_i, (t+1) \cdot \frac{1}{d_i} + r_i)$ clearly contains at most one integral point. The event that $\{[t \cdot \frac{1}{d_i} + r_i, (t+1) \cdot \frac{1}{d_i} + r_i) \cap \mathbb{N} \neq \emptyset\}$ is equivalent to the event that a continuous window of size equal to $\frac{1}{d_i}$ starting from the (real) point $t \cdot \frac{1}{d_i} + r_i$ contains an integer. For $r_i$ ranging in $[0, 1]$, the starting point of the interval lies between $t \cdot \frac{1}{d_i}$ and $t \cdot \frac{1}{d_i} + 1$. It is not hard to see that fraction of possible realizations of $r_i$ such that the window contains an integer equals its size. The fact follows since for any $i \in \mathcal{A}$, the window has size $\frac{1}{d_i}$ and the offset $r_i$ is sampled uniformly at random from $[0, 1]$. $\qquad\square$

**Lemma 3.3.** *Let $\mathcal{R}^{CP}(T)$ be the optimal solution to* (**CP**) *and* $\mathrm{OPT}(T)$ *be the optimal solution over $T$ rounds. We have $\mathcal{R}^{CP}(T) \geq \mathrm{OPT}(T) - \mathcal{O}(d_{\max} f(\mathcal{A}))$, where $d_{\max} = \max_{i \in \mathcal{A}}\{d_i\}$.*

*Proof.* In order to prove the lemma, we first construct an (non-convex) IP upper bound on the optimal expected reward over $T$ rounds, based on the multi-linear extension of $f$.

$$\textbf{maximize:} \quad \sum_{i \in [T]} \sum_{S \subseteq \mathcal{A}} f(S) \prod_{i \in S} x_{i,t} \prod_{i \notin S} (1 - x_{i,t}) \qquad\qquad \textbf{(MP)}$$

$$\textbf{s.t.} \quad \sum_{t' \in [t, t+d_i-1]} x_{i,t'} \leq 1, \forall i \in \mathcal{A}, \forall t \in [T] \qquad\qquad (4)$$

$$\mathbf{x}_t \in \{0, 1\}^k, \forall t \in [T]$$

In the formulation (**MP**), each variable $x_{i,t}$ can be thought of as the 0-1 indicator of playing arm $i \in \mathcal{A}$ at time $t \in [T]$. Intuitively, constraints (4) of (**MP**) indicate the fact that, due to blocking constraints, each arm $i \in \mathcal{A}$ can be played at most once every $d_i$ steps. Clearly, any optimal solution to RSW can be mapped onto the above formulation and, thus, the optimal solution of (**MP**) provides an upper bound on $\mathrm{OPT}(T)$.

Let $\mathbf{x}_t \in \{0, 1\}^k$ for each $t \in [T]$ be a vector such that $(\mathbf{x}_t)_i = x_{i,t}$. Notice that for any integral $\mathbf{x} \in \{0, 1\}^k$, the multi-linear extension is equal to the concave closure of any set function $f$, that is, $f^+(\mathbf{x}) = F(\mathbf{x})$. Therefore, (**MP**) remains an upper bound, even if we replace its objective function with $g(\mathbf{x}_1, \ldots, \mathbf{x}_T) = \sum_{t \in [T]} f^+(\mathbf{x}_t)$.

We now fix any optimal solution $\{x_{i,t}^*\}_{i \in \mathcal{A}, t \in [T]}$ to (**MP**) under the objective $g(\mathbf{x}_1, \ldots, \mathbf{x}_T) = \sum_{t \in [T]} f^+(\mathbf{x}_t)$. Let us define the variables $\{z_i'\}_{i \in \mathcal{A}}$, such that

$$z_i' = \frac{1}{T} \sum_{t \in [T]} x_{i,t}^* \geq 0, \quad \forall i \in \mathcal{A}.$$

In the above definition, each $z_i'$ is the fraction of time an element $i \in \mathcal{A}$ is chosen in an optimal solution. Let $\mathbf{z}' \in [0, 1]^k$, such that $(\mathbf{z}')_i = z_i' \; \forall i \in \mathcal{A}$.

By concavity of $f^+$, we have

$$g(\mathbf{x}_1^*, \ldots, \mathbf{x}_T^*) = \sum_{t \in [T]} f^+(\mathbf{x}_t^*) = T \sum_{t \in [T]} \frac{1}{T} f^+(\mathbf{x}_t^*) \leq T f^+\left(\frac{1}{T} \sum_{t \in [T]} \mathbf{x}_t^*\right) = T f^+(\mathbf{z}'),$$

where the inequality follows by the fact that $\mathbf{z}'$ can be thought of as a convex combination of $\{\mathbf{x}_1^*, \ldots, \mathbf{x}_T^*\}$.

Moreover, for each $i \in \mathcal{A}$ and by averaging constraints (4) of (**MP**) over all $t \in [T]$, we can see that

$$\frac{1}{T} \sum_{t \in [1, d_i-1]} t x_{i,t}^* + \frac{1}{T} \sum_{t \in [d_i, T]} d_i x_{i,t}^* \leq 1 \Leftrightarrow \frac{1}{T} \sum_{t \in [T]} d_i x_{i,t}^* \leq 1 + \frac{1}{T} \sum_{t \in [1, d_i-1]} (d_i - t) x_{i,t}^*.$$

Given the fact that $\sum_{t \in [1, d_i-1]} x_{i,j}^* \leq 1$, the above inequality immediately implies that

$$z_i' \leq \frac{1}{d_i}\left(1 + \frac{d_i - 1}{T}\right) \quad \forall i \in \mathcal{A}.$$

Consider now the assignment $z_i = \left(1 + \frac{d_{\max}-1}{T}\right)^{-1} z'_i$, $\forall i \in \mathcal{A}$. For this assignment, we can easily verify that the constraints of (**CP**) are trivially satisfied, since $0 \leq z_i \leq \frac{1}{d_i}$, $\forall i \in \mathcal{A}$.

Let $\mathbf{z} \in [0,1]^k$, such that $(\mathbf{z})_i = z_i$ $\forall i \in \mathcal{A}$. By the above analysis, we can see that

$$\mathbf{z} = \mathbf{z}' - \frac{d_{\max}-1}{T + d_{\max} - 1} \mathbf{z}',$$

where we use the fact that $\frac{1}{1+\beta} = 1 - \frac{\beta}{1+\beta}$ for any $\beta \in \mathbb{R}$. Finally, by concavity of $f^+$ we have

$$
\begin{aligned}
f^+(\mathbf{z}) = f^+ &\left( \left(1 - \frac{d_{\max}-1}{T + d_{\max} - 1}\right) \mathbf{z}' + \frac{d_{\max}-1}{T + d_{\max} - 1} \mathbf{0}\right) \\
&\geq \left(1 - \frac{d_{\max}-1}{T + d_{\max} - 1}\right) f^+(\mathbf{z}') + \frac{d_{\max}-1}{T + d_{\max} - 1} f^+(\mathbf{0}) \\
&\geq f^+(\mathbf{z}') - \frac{d_{\max}-1}{T + d_{\max} - 1} f(\mathcal{A}),
\end{aligned}
$$

where the last inequality follows by the facts that $f^+(\mathbf{0}) = f(\mathbf{0}) = 0$ and $f^+(\mathbf{z}') \leq f^+(\mathbf{1}) = f(\mathcal{A})$, since $f$ is monotone.

Therefore, by exhibiting a feasible solution $\mathbf{z}$ of (**CP**) such that

$$T f^+(\mathbf{z}) \geq T f^+(\mathbf{z}') - \mathcal{O}(d_{\max} f(\mathcal{A})) \geq g(\mathbf{x}_1^*, \ldots, \mathbf{x}_T^*) - \mathcal{O}(d_{\max} f(\mathcal{A})) \geq \mathrm{OPT}(T) - \mathcal{O}(d_{\max} f(\mathcal{A})),$$

the proof is completed. $\square$

## C.2 Hardness of approximation

The goal of this section is to show that the $\left(1 - \frac{1}{e}\right)$-multiplicative factor in the approximation guarantee of Theorem 1.3 cannot be improved, unless $\mathbf{P} = \mathbf{NP}$. Specifically, we prove the following result:

**Theorem 3.5.** *For any $\epsilon > 0$, there exists no polynomial-time $\left(1 - \frac{1}{e} + \epsilon\right)$-approximation algorithm for the RSW problem, unless $\mathbf{P} = \mathbf{NP}$, even in the special case where $d_{\max} = o(T)$.*

In order show the above hardness result, we study for simplicity the average version of RSW, where the objective is to maximize the average reward over $T$ time steps, namely, $\frac{1}{T}\left(\sum_{t \in [T]} f(\mathcal{A}_t)\right)$, where $\mathcal{A}_t$ is the set of elements used at time $t \in [T]$. Notice that in the average case, the additive term in the approximation guarantee of INTERLEAVED-GREEDY, as presented in Theorem 1.3, vanishes as $T \to \infty$. Let OPT be the average reward collected by any optimal algorithm for RSW.

Our proof relies on a reduction from the Submodular Welfare (SW) problem [47], in the special case where the players have identical utility functions. The problem can be formally defined as follows:

**Definition C.1** (Submodular Welfare with Identical Utilities (SWIU)). *We consider a set of $k$ items and $m$ players, each associated with the same monotone submodular utility function $u : 2^{[k]} \to \mathbb{R}_{\geq 0}$ over the items. The goal is to partition the $k$ items into $m$ subsets $S_1, \ldots, S_m$, such that to maximize $\sum_{i \in [m]} u(S_i)$.*

As noted in [47], the hardness result presented in [26] for the SW problem also holds for SWIU, namely, the special case of SW where all the players have the same utility function. Note, also that the RSW problem is defined in the *value oracle* model, as we are only allowed to make queries of the function value for any input set.

**Theorem C.2** ([26]). *For any $\epsilon > 0$, there exists no polynomial-time $\left(1 - \frac{1}{e} + \epsilon\right)$-approximation algorithm for the SWIU problem in the value oracle model, unless $\mathbf{P} = \mathbf{NP}$.*

We start from a simple construction for the non-average case of RSW in order to show how our problem is directly associated with SWIU: Consider an instance of SWIU of $k$ items and $m$ players. Let $u : 2^{[k]} \to \mathbb{R}_{\geq 0}$ be the monotone submodular utility function which is commonly used by all players. Given the above instance, we can construct in polynomial time an instance of RSW as follows: Let $\mathcal{A}$ be the set of $k$ elements, each corresponding to an item, and let $f : 2^{\mathcal{A}} \to \mathbb{R}_{\geq 0}$ be our

function, chosen such that $f \equiv u$. We set the delay of each element $i \in \mathcal{A}$ as well as the time horizon to be equal to the number of players, namely, $d_i = T = m$ for each $i \in \mathcal{A}$.

Clearly, in the above construction where the delays are all equal to the time horizon, each element can be chosen at most once by any algorithm for RSW. Therefore, the above constructed instance of RSW exactly corresponds to SWIU, given that any solution to latter immediately translates into a solution of RSW of the same total reward, and the opposite.

The above construction immediately relates the two problems in the case where the delays can be of the same order as the time horizon. However, it does not rule out the possibility that the RSW problem might become easier in the special case where $d_{\max} = o(T)$. Indeed, one could argue that for small enough delays, exploiting the possible periodicity of the RSW solutions might lead to improved approximation guarantees. Notice, further, that the approximation guarantee we provide in Theorem 1.3 for IS becomes meaningless in the above scenario, since the additive loss for $d_{\max} = T$ becomes $\mathcal{O}(T \cdot f(\mathcal{A}))$.

In order to overcome the above technical issue and show that the multiplicative factor of $\left(1 - \frac{1}{e}\right)$ in Theorem 1.3 cannot be improved, we map any instance of SWIU onto an instance of RSW such that $T \gg d_{\max}$. Given any instance of SWIU, we can construct in polynomial time an instance of RSW as follows: We define $\mathcal{A}$ to be the set of $k$ items, $f \equiv u$ to be the monotone submodular function and $d_i = m \ \forall i \in \mathcal{A}$ to be the delay of all elements. In this case, we consider a time horizon $T = m \cdot \lceil \text{poly}(k, m) \rceil$, where by $\text{poly}(k, m)$ we denote some polynomial function in $k$ and $m$.

We first show that, without loss of generality, we can focus our attention on solutions to the average case of RSW that exhibit a periodic structure of period $m$.

**Lemma C.3.** *Let $\nu : [T] \to 2^{\mathcal{A}}$ be any feasible assignment to the above instance of RSW of average reward $R$. We can construct in polynomial time a feasible assignment $\nu' : [T] \to 2^{\mathcal{A}}$ of average reward at least $R' \geq R$, such that $\nu'(t) = \nu(t + m) \ \forall t \in \mathbb{N}$, namely, $\nu'$ is a periodic assignment of period $m$.*

*Proof.* Given that the average reward of the assignment $\nu$ is $R$, there must exist a continuous subsequence of rounds of length $m$, that is, $\{t, \ldots, t + m - 1\}$ for some $t \in [T - m]$, such that

$$\frac{1}{m} \sum_{\tau = t}^{t + m - 1} f(\nu(t)) \geq R.$$

In the opposite case, we immediately get a contradiction to the fact that the average reward is at least $R$.

Let $L$ with $|L| = m$ be such a sequence. We now construct the periodic assignment $\nu'$ by repeating the assignment of the subinterval $L$, as follows:

$$\nu'(t) = \nu(L(t \mod m)) \in 2^{\mathcal{A}} \ \forall t \in [T].$$

It is not hard to verify that since $d_i = m$ for each $i \in \mathcal{A}$ and since $L$ is a subsequence of a feasible assignment of length $m$, the assignment $\nu'$ never violates the blocking constraints. Moreover, the average reward of $\nu'$ equals the average reward of the interval $L$ which is at least $R$. Finally, notice that the subsequence $L$ can be found in polynomial time, given the fact that the time horizon $T$ is defined to be polynomial in $k$ and $m$. $\qquad \square$

We can now complete the proof of our hardness result.

*Proof of Theorem 3.5.* We prove the result via a reduction from the SWIU problem to the average version of the RSW. Clearly, the average and non-average version of RSW share the same approximability status, as the two problems are essentially identical up to a scaling of the objective function.

Given an instance $I$ of SWIU, we can construct in polynomial time an instance $I'$ of the average version of RSW, as described above. Let $\text{OPT}_{SWIU}(I)$ and $\text{OPT}_{RSW}(I')$ be the optimal solution of SWIU and RSW on the corresponding instance, respectively.

We first show that when $\text{OPT}_{SWIU}(I) \geq R$ for some reward $R$, then we necessarily have that $\text{OPT}_{RSW}(I') \geq \frac{R}{m}$. Indeed, let $L : [m] \to 2^{[k]}$ be an allocation that achieves a reward $R' =$

$\mathrm{OPT}_{SWIU}(I) \geq R$ for the instance $I$ of SWIU. As indicated in proof of Lemma C.3, we can construct in polynomial time a periodic assignment for the RSW problem of average reward exactly $\frac{R'}{m}$, which implies that $\mathrm{OPT}_{RSW}(I') \geq \frac{R'}{m} \geq \frac{R}{m}$.

Now, we would like to show that if $\mathrm{OPT}_{SWIU}(I) \leq \alpha R$ for some reward $R$ and $\alpha \in (0,1)$, then it has to be that $\mathrm{OPT}_{RSW}(I') \leq \alpha \frac{R}{m}$. We prove the statement via its contrapositive, assuming that $\mathrm{OPT}_{RSW}(I') > \alpha \frac{R}{m}$ for some reward $R$ and $\alpha \in (0,1)$. Let $\frac{R'}{m} > \alpha \frac{R}{m}$ be the optimal average reward of RSW. By Lemma C.3, we can assume w.l.o.g. that the assignment $\mathrm{OPT}_{RSW}(I')$, that achieves an average reward of $\frac{R'}{m}$, is a periodic assignment of period $m$. However, given that all the delays are equal to $m$ in the instance $I'$ of RSW, it is easy to see that in any period of $m$ consecutive rounds, each element is played at most once. Moreover, the average reward of each period is exactly $\frac{R'}{m}$. Therefore, any continuous subsequence of length $m$ in the solution of the RSW naturally induces a solution to the instance $I$ of SWIU of total reward exactly $R'$. This, in turn, implies that $\mathrm{OPT}_{SWIU}(I) \geq R' \geq \alpha R$.

By the above discussion, we have completed the proof of a reduction from SWIU to RSW. Therefore, any polynomial-time $\left(1 - \frac{1}{e} + \epsilon\right)$-approximation algorithm for RSW, for some $\epsilon > 0$, would imply a $\left(1 - \frac{1}{e} + \epsilon\right)$-approximation algorithm for SWIU. However, by Theorem C.2 this is not possible, unless $\mathbf{P} = \mathbf{NP}$. $\qquad\square$

We believe that, through a similar reduction as above, we can prove information-theoretic hardness of the RSW problem by leveraging the results in [36]. We leave this as future work.

# D  Matroid Blocking Semi-Bandits: Omitted Proofs

**Theorem 4.2.** *The expected reward collected by* INTERLEAVED-GREEDY *for $T$ rounds, $\mathcal{R}^{IG}(T)$, is at least $\left(1 - \frac{1}{e}\right) \mathrm{OPT}(T) - \mathcal{O}(d_{\max} \mathrm{rk}(\mathcal{M}))$, where $\mathrm{OPT}(T)$ is the optimal expected reward.*

*Proof.* Fix any algorithm for the MBB problem and let $\mathcal{A}_t$ be the set of arms played at round $t$. Notice that the sets $\{\mathcal{A}_t\}_{t \in [T]}$ are independent of the reward realizations, since the selection of arms pulled at each round is made before observing their actual rewards. Thus, the expected reward collected (over the randomness of the reward realizations) can be expressed as

$$\mathbb{E}\left[\sum_{t \in [T]} \sum_{i \in \mathcal{A}_t} X_{i,t}\right] = \sum_{t \in [T]} \sum_{i \in \mathcal{A}_t} \mathbb{E}\left[X_{i,t}\right] = \sum_{t \in [T]} \sum_{i \in \mathcal{A}_t} \mu_i.$$

Therefore, INTERLEAVED-GREEDY can be thought of as an instance of INTERLEAVED-SUBMODULAR for the weighted rank function of the given matroid, that is, for $f_{\mathcal{M},\mu}(S) = \max_{I \in \mathcal{I}, I \subseteq S}\{\mu(I)\}$. By Lemma 2.3, this function is monotone submodular and, also, $f_{\mathcal{M},\mu}(\mathcal{A}) \leq \mathrm{rk}(\mathcal{M})$, given that the distribution of rewards is bounded in $[0,1]$.

Thus, by applying Theorem 1.3, we can conclude that

$$\mathcal{R}^{IG}(T) \geq \left(1 - \frac{1}{e}\right) \mathcal{R}^{LP}(T) \geq \left(1 - \frac{1}{e}\right) \mathrm{OPT}(T) - \mathcal{O}(d_{\max} \mathrm{rk}(\mathcal{M})).$$

$\qquad\square$

**Lemma 4.4.** *Let $\bar{\mu}_t(S) = \sum_{i \in S} \bar{\mu}_{i,t}$ and $\mu(S) = \sum_{i \in S} \mu_i$. We have*

$$\mathcal{R}^{IG}(T) - \mathcal{R}^{IB}(T) = \mathbb{E}_{\mathbf{r} \sim U[0,1]^k, \mathcal{Q}}\left[\sum_{t \in [T]}\left(\max_{S \subseteq \mathcal{G}_t(\mathbf{r}), S \in \mathcal{I}}\{\mu(S)\} - \mu\left(\arg\max_{S \subseteq \mathcal{G}_t(\mathbf{r}), S \in \mathcal{I}}\{\bar{\mu}_t(S)\}\right)\right)\right].$$

*Proof.* Let $\{\mathcal{G}_t(\mathbf{r})\}_{t \in [T]}$ be the sequence of sampled arms over $T$ rounds as a function of the sampled offsets $\mathbf{r} \in [0,1]^k$. Moreover, let $X_t(S)$ be the realized rewards of a subset $S \subseteq \mathcal{A}$ of arms at round $t \in [T]$. We denote by $\mathcal{A}_t^\pi$ the arms played at round $t \in [T]$ and by $H_t^\pi = \{\mathcal{A}_1^\pi, X_1(\mathcal{A}_1^\pi), \ldots, \mathcal{A}_t^\pi, X_t(\mathcal{A}_t^\pi)\}$ the *history* of arm playing and observed realizations up to (and

including) time $t$ by algorithm $\pi \in \{IG, IB\}$. Recall that we denote by $\mathcal{Q}$ the randomness due to the reward realizations of the arms.

Notice that in the case of IB and for fixed offsets, the player's actions only depend on the previous realized rewards of the arms. Thus, for any fixed offset vector $\mathbf{r}^{IB}$, we have

$$\mathbb{E}_{\mathcal{Q}} \left[ \sum_{i \in \mathcal{A}} X_{i,t} \, \mathcal{X} \left( i \in \arg \max_{S \subseteq \mathcal{G}_t(\mathbf{r}^{IB}), S \in \mathcal{I}} \{\bar{\mu}_t(S)\} \right) \right]$$

$$= \mathbb{E}_{\mathcal{Q}} \left[ \sum_{i \in \mathcal{A}} \mathbb{E}_{\mathcal{Q}} \left[ X_{i,t} \, \mathcal{X} \left( i \in \arg \max_{S \subseteq \mathcal{G}_t(\mathbf{r}^{IB}), S \in \mathcal{I}} \{\bar{\mu}_t(S)\} \right) \mid H_{t-1}^{IB} \right] \right]$$

$$= \mathbb{E}_{\mathcal{Q}} \left[ \sum_{i \in \mathcal{A}} \mathbb{E}_{\mathcal{Q}} \left[ X_{i,t} \mid H_{t-1}^{IB} \right] \mathcal{X} \left( i \in \arg \max_{S \subseteq \mathcal{G}_t(\mathbf{r}^{IB}), S \in \mathcal{I}} \{\bar{\mu}_t(S)\} \right) \right]$$

$$= \mathbb{E}_{\mathcal{Q}} \left[ \sum_{i \in \mathcal{A}} \mu_i \, \mathcal{X} \left( i \in \arg \max_{S \subseteq \mathcal{G}_t(\mathbf{r}^{IB}), S \in \mathcal{I}} \{\bar{\mu}_t(S)\} \right) \right]$$

$$= \mathbb{E}_{\mathcal{Q}} \left[ \mu \left( \arg \max_{S \subseteq \mathcal{G}_t(\mathbf{r}^{IB}), S \in \mathcal{I}} \{\bar{\mu}_t(S)\} \right) \right].$$

Similarly, notice that the algorithm IG is oblivious to the realized rewards. Therefore, for any fixed offset vector $\mathbf{r}^{IG}$ and at any time $t \in [T]$, we get

$$\mathbb{E}_{\mathcal{Q}} \left[ \sum_{i \in \mathcal{A}} X_{i,t} \, \mathcal{X} \left( i \in \arg \max_{S \subseteq \mathcal{G}_t(\mathbf{r}^{IG}), S \in \mathcal{I}} \{\mu(S)\} \right) \right] = \mathbb{E}_{\mathcal{Q}} \left[ \sum_{i \in \mathcal{A}} \mu_i \, \mathcal{X} \left( i \in \arg \max_{S \subseteq \mathcal{G}_t(\mathbf{r}^{IG}), S \in \mathcal{I}} \{\mu(S)\} \right) \right]$$

$$= \mathbb{E}_{\mathcal{Q}} \left[ \max_{S \subseteq \mathcal{G}_t(\mathbf{r}^{IG}), S \in \mathcal{I}} \{\mu(S)\} \right].$$

The lemma follows by observing that the offsets $\mathbf{r}^{IG}$ and $\mathbf{r}^{IB}$ of the two algorithms follow exactly the same distribution. Therefore, we have

$$\mathcal{R}^{IG}(T) - \mathcal{R}^{IB}(T)$$

$$= \mathbb{E}_{\mathbf{r}^{IG} \sim [0,1]^k, \mathcal{Q}} \left[ \sum_{t \in [T]} \max_{S \subseteq \mathcal{G}_t(\mathbf{r}^{IG}), S \in \mathcal{I}} \{\mu(S)\} \right] - \mathbb{E}_{\mathbf{r}^{IB} \sim [0,1]^k, \mathcal{Q}} \left[ \sum_{t \in [T]} \mu \left( \arg \max_{S \subseteq \mathcal{G}_t(\mathbf{r}^{IB}), S \in \mathcal{I}} \{\bar{\mu}_t(S)\} \right) \right]$$

$$= \mathbb{E}_{\mathbf{r} \sim [0,1]^k, \mathcal{Q}} \left[ \sum_{t \in [T]} \left( \max_{S \subseteq \mathcal{G}_t(\mathbf{r}), S \in \mathcal{I}} \{\mu(S)\} - \mu \left( \arg \max_{S \subseteq \mathcal{G}_t(\mathbf{r}), S \in \mathcal{I}} \{\bar{\mu}_t(S)\} \right) \right) \right].$$

$\square$

**Lemma 4.7.** *Under the event $\mathcal{E}_{\mathbf{r}}$ and at any time $t \in [T]$, we have $\Delta_{\mathcal{G}_t}(\bar{\mu}_t) = \sum_{i \in \mathcal{A}_t^{IG}} \Delta_{i, \sigma_t^{-1}(i)}$.*

*Proof.* Recall that under the event $\mathcal{E}_{\mathbf{r}}$, both algorithms IG and IB use the same offset vector $\mathbf{r}$ and, thus, they operate on same sequence of sampled arms over time. Let $\mathcal{G}_t = \mathcal{G}_t(\mathbf{r})$ be the common set of sampled arms and let $\mathcal{A}_t^{IG}$ and $\mathcal{A}_t^{IB}$ be the maximal independent sets computed by IG and IB, respectively, at any round $t \in [T]$. Notice that for any $t \in [T]$ both $\mathcal{A}_t^{IG}$ and $\mathcal{A}_t^{IB}$ are bases of the restricted matroid $\mathcal{M} \mid \mathcal{G}_t$ and, thus, correspond to independent sets of $\mathcal{I}$ of equal cardinality. Let $\sigma_t$ be the bijection between $\mathcal{A}_t^{IG}$ and $\mathcal{A}_t^{IB}$ described by Theorem 4.6. For any $t \in [T]$, we have that

$$\Delta_{\mathcal{G}_t}(\bar{\mu}) = \mu(\mathcal{A}_t^{IG}) - \mu(\mathcal{A}_t^{IB}) = \sum_{i \in \mathcal{A}_t^{IG}} \mu_i - \sum_{j \in \mathcal{A}_t^{IB}} \mu_j = \sum_{i \in \mathcal{A}_t^{IG}} \left( \mu_i - \mu_{\sigma_t^{-1}(i)} \right) = \sum_{i \in \mathcal{A}_t^{IG}} \Delta_{i, \sigma_t^{-1}(i)}.$$

$\square$

**Lemma 4.8.** *Let* $\ell_{i,j} = \left\lfloor \frac{8\ln(T)}{\Delta_{i,j}^2} \right\rfloor$ *for any $i < j$. Under event $\mathcal{E}_{\mathbf{r}}$ and for any arm $j > 1$, we have*

$$\sum_{t\in[T]}\sum_{i<j}\Delta_{i,j}\,\mathcal{X}\left(\sigma_t(j)=i, T_j(t)\leq\ell_{i,j}\right) \leq \frac{16}{\Delta_{j-1,j}}\ln(T) \qquad \textit{(Under-sampled regret)} \quad (2)$$

$$\mathbb{E}_{\mathcal{Q}}\left[\sum_{t\in[T]}\sum_{i<j}\Delta_{i,j}\,\mathcal{X}\left(\sigma_t(j)=i, T_j(t)>\ell_{i,j}\right)\right] \leq \frac{\pi^2}{3}\sum_{i=1}^{j-1}\Delta_{i,j} \quad \textit{(Sufficiently sampled regret)} \quad (3)$$

*Proof.* We first focus on proving inequality (2), that is, the part of the regret attributed to an arm $j > 1$ when not enough samples have been collected. Notice that the algorithm $IB$ never accumulates regret when it plays the arm $j = 1$ of highest mean reward. Recall that for any fixed $j \in \mathcal{A}$, we have $\Delta_{1,j} > \Delta_{2,j} > \cdots > \Delta_{j,j} = 0$, since we assume w.l.o.g. that the arms have distinct mean rewards. By construction of our algorithm, if the number of samples from arm $j \in \mathcal{A}$ is increased at some round $t$, it is because there exists exactly one arm $i \in \mathcal{A}$ with $\Delta_{i,j} > 0$, such that $\sigma_t(j) = i$. The above is implied by Theorem 4.6, given the fact that each bijection $\sigma_t$ for all $t \in [T]$ maps each arm played by IB in $\mathcal{A}_t^{IB}$ to a single arm played by IG in $\mathcal{A}_t^{IG}$. On the other hand, as the number of obtained samples $T_j(t)$ from arm $j \in \mathcal{A}$ by time $t \in [T]$ increases, the maximum suboptimality gap $\Delta_{i,j}$ that can be charged in the under-sampled part of the regret is that of the maximum reward $i \in \mathcal{A}$ that satisfies $T_j(t) \leq \ell_{i,j}$. By the above analysis, for any $j > 1$, we get that

$$\sum_{t\in[T]}\sum_{i=1}^{j-1}\Delta_{i,j}\,\mathcal{X}\left(\sigma_t(j)=i, T_j(t)\leq\ell_{i,j}\right) \leq \sum_{i=1}^{j-1}\left(\Delta_{i,j}-\Delta_{i+1,j}\right)\ell_{i,j}$$

$$\leq \sum_{i=1}^{j-1}\left(\Delta_{i,j}-\Delta_{i+1,j}\right)\frac{8\ln(T)}{\Delta_{i,j}^2}, \qquad (5)$$

where the last inequality follows by definition of $\ell_{i,j}$.

The rest of the claim follows by simple algebra. Indeed,

$$(5) \leq \left(\sum_{i=1}^{j-1}\frac{\Delta_{i,j}-\Delta_{i+1,j}}{\Delta_{i,j}^2}\right)8\ln(T)$$

$$\leq \left(\frac{1}{\Delta_{j-1,j}}+\sum_{i=1}^{j-2}\frac{\Delta_{i,j}-\Delta_{i+1,j}}{\Delta_{i,j}^2}\right)8\ln(T)$$

$$\leq \left(\frac{1}{\Delta_{j-1,j}}+\sum_{i=1}^{j-2}\frac{\Delta_{i,j}-\Delta_{i+1,j}}{\Delta_{i,j}\Delta_{i+1,j}}\right)8\ln(T)$$

$$= \left(\frac{1}{\Delta_{j-1,j}}+\sum_{i=1}^{j-2}\left(\frac{1}{\Delta_{i+1,j}}-\frac{1}{\Delta_{i,j}}\right)\right)8\ln(T)$$

$$= \left(\frac{2}{\Delta_{j-1,j}}-\frac{1}{\Delta_{1,j}}\right)8\ln(T)$$

$$\leq \frac{16}{\Delta_{j-1,j}}\ln(T).$$

We now focus on proving inequality (3), that is, the regret accumulated after a sufficient number of samples has been collected from an arm $j > 1$. Notice, that given the event $\mathcal{E}_{\mathbf{r}}$, the expectation in the LHS of inequality (3) is taken only over the randomness of the realized rewards that are observed by IB.

For proving the upper bound, we fix any arm $j > 1$ and focus on each arm $i \in \mathcal{A}$ such that $i < j$ and, thus, $\Delta_{i,j} > 0$. Let us fix any such arm $i \in \mathcal{A}$. For any $t \in [T]$, the event $\{\sigma_t(j) = i\}$ implies that $\{\mu_i > \mu_j, \bar{\mu}_{i,t} \leq \bar{\mu}_{j,t}\}$, namely, the order of the UCB-indices at time $t \in [T]$ of $i$ and $j$ is inconsistent with the order of their true mean rewards. In the opposite case, the algorithm IB would

have chosen the set $\mathcal{A}_t^{IB} - j + i$, which, as suggested by Theorem 4.6, is an independent set of $\mathcal{M}$. Therefore, for any arm $i < j$, we have

$$\{\sigma_t(j) = i, T_j(t) > \ell_{i,j}\} \subseteq \{\bar{\mu}_{i,t} \leq \bar{\mu}_{j,t}, \mu_i > \mu_j, T_j(t) > \ell_{i,j}\}. \tag{6}$$

Note that the inclusion in the above expression is because the inconsistency in the order of UCB-indices does not necessarily imply that $\sigma_t(j) = i$ (i.e., that IB actually exchanges $j$ for $i$ at time $t \in [T]$).

By definition of the UCB-indices, the event $\bar{\mu}_{i,t} \leq \bar{\mu}_{j,t}$ at time $t \in [T]$ implies that

$$\hat{\mu}_{i,T_i(t)} + \sqrt{\frac{2\ln(t)}{T_i(t)}} \leq \hat{\mu}_{j,T_j(t)} + \sqrt{\frac{2\ln(t)}{T_j(t)}}. \tag{7}$$

We fix $s_i = T_i(t)$ and $s_j = T_j(t) > \ell_{i,j}$ to be the number of samples obtained from arm $i$ and $j$, respectively, by time $t \in [T]$. Notice that in order for (7) to hold, at least one of the following events must be true:

**(i)** $\left\{\hat{\mu}_{i,s_i} + \sqrt{\frac{2\ln(t)}{s_i}} \leq \mu_i\right\}$, **(ii)** $\left\{\hat{\mu}_{j,s_j} \geq \mu_j + \sqrt{\frac{2\ln(t)}{s_j}}\right\}$, **(iii)** $\left\{\mu_i < \mu_j + 2\sqrt{\frac{2\ln(t)}{s_j}}\right\}$.

Indeed, it can be easily verified that the simultaneous negation of the above three events contradicts (7) for any fixed number of samples $s_i, s_j$.

By our choice of $\ell_{i,j} = \left\lfloor \frac{8\ln(T)}{\Delta_{i,j}^2} \right\rfloor$ and the fact that $s_j \geq \ell_{i,j} + 1 \geq \frac{8\ln(T)}{\Delta_{i,j}^2}$, we can see that event **(iii)** cannot be true, since in that case, we have

$$\mu_j + 2\sqrt{\frac{2\ln(t)}{s_j}} \leq \mu_j + 2\sqrt{\frac{2\Delta_{i,j}^2\ln(t)}{8\ln(T)}} \leq \mu_j + \Delta_{i,j} = \mu_i.$$

Moreover, by Hoeffding's inequality, for the probabilities of the events **(i)** and **(ii)**, we have that

$$\mathbb{P}\left(\hat{\mu}_{i,s_i} + \sqrt{\frac{2\ln(t)}{s_i}} \leq \mu_i\right) \leq e^{-4\ln(t)} = t^{-4} \text{ and } \mathbb{P}\left(\hat{\mu}_{j,s_j} \geq \mu_j + \sqrt{\frac{2\ln(t)}{s_j}}\right) \leq e^{-4\ln(t)} = t^{-4},$$

where the probability is taken over the randomness of the reward realizations.

Therefore, for any numbers of samples $s_i = T_i(t)$ and $s_j = T_j(t) > \ell_{i,j}$, we have

$$\mathbb{P}\left(\bar{\mu}_{i,t} \leq \bar{\mu}_{j,t}, \mu_i > \mu_j, T_j(t) = s_j, T_i(t) = s_i\right) \leq \mathbb{P}\left(\hat{\mu}_{i,s_i} + \sqrt{\frac{2\ln(t)}{s_i}} \leq \mu_i\right) + \mathbb{P}\left(\hat{\mu}_{j,s_j} \geq \mu_j + \sqrt{\frac{2\ln(t)}{s_j}}\right)$$

$$\leq 2 \cdot t^{-4}. \tag{8}$$

Finally, by union bound over the possible number of samples, $s_i$ and $s_j$, and using the aforementioned results, for any $j > 1$ and time $t \in [T]$, we have

$$\mathbb{E}_{\mathcal{Q}}\left[\sum_{t \in [T]} \sum_{i=1}^{j-1} \Delta_{i,j} \mathcal{X}\left(\sigma_t(j) = i, T_j(t) > \ell_{i,j}\right)\right]$$

$$= \mathbb{E}_{\mathcal{Q}}\left[\sum_{t \in [T]} \sum_{i=1}^{j-1} \sum_{s_i=0}^{t-1} \sum_{s_j=\ell_{i,j}+1}^{t-1} \Delta_{i,j} \mathcal{X}\left(\sigma_t(j) = i, T_j(t) = s_j, T_i(t) = s_i\right)\right] \tag{9}$$

$$\leq \mathbb{E}_{\mathcal{Q}}\left[\sum_{t \in [T]} \sum_{i=1}^{j-1} \sum_{s_i=0}^{t-1} \sum_{s_j=\ell_{i,j}+1}^{t-1} \Delta_{i,j} \mathcal{X}\left(\bar{\mu}_{i,t} \leq \bar{\mu}_{j,t}, \mu_i > \mu_j, T_j(t) = s_j, T_i(t) = s_i\right)\right] \tag{10}$$

$$= \sum_{t \in [T]} \sum_{i=1}^{j-1} \sum_{s_i=0}^{t-1} \sum_{s_j=\ell_{i,j}+1}^{t-1} \Delta_{i,j} \mathbb{P}\left(\bar{\mu}_{i,t} \leq \bar{\mu}_{j,t}, \mu_i > \mu_j, T_j(t) = s_j, T_i(t) = s_i\right)$$

$$\leq \sum_{t \in [T]} \sum_{i=1}^{j-1} \Delta_{i,j} 2t(t-1)t^{-4}, \tag{11}$$

where in (9) we consider any possible number of samples by time $t$ for each arm. Moreover, inequality (10) follows by (6) and (11) follows by (8). The proof of inequality (3) follows by the fact that

$$\sum_{t \in [T]} t(t-1)t^{-4} \leq \sum_{t \in [T]} t^{-2} \leq \sum_{t=1}^{+\infty} t^{-2} = \frac{\pi^2}{6}.$$

$\square$

## D.1 Proof of Theorem 1.4

**Theorem 1.4.** *The expected reward collected by* INTERLEAVED-UCB *in* $T$ *rounds,* $\mathcal{R}^{IB}(T)$, *for* $k$ *arms, a matroid of rank* $r = \mathrm{rk}(\mathcal{M})$ *and maximum delay* $d_{\max}$ *is at least*

$$\left(1 - \frac{1}{e}\right) \mathrm{OPT}(T) - \mathcal{O}\left(k\sqrt{T \ln(T)} + k^2 + d_{\max}r\right).$$

*Proof.* By inequality (1), Lemma 4.4 and Definition 4.5, we can upper bound the $\alpha$-regret, for $\alpha = 1 - \frac{1}{e}$, as

$$\alpha \mathrm{OPT}(T) - \mathcal{R}^{IB}(T) \leq \mathop{\mathbb{E}}_{\mathbf{r} \sim [0,1]^k, \mathcal{Q}} \left[ \sum_{t \in [T]} \Delta_{\mathcal{G}_t(\mathbf{r})}(\bar{\mu}_t) \right] + \mathcal{O}(d_{\max} \cdot \mathrm{rk}(\mathcal{M})), \qquad (12)$$

where the expectation is taken over the randomness of the offset vector $\mathbf{r}$ and the reward realizations.

Under the event $\mathcal{E}_{\mathbf{r}}$, that is, where both IG and IB use the same offsets $\mathbf{r}$, let $\{\sigma_t\}_{t \in [T]}$ be the sequence of bijections between $\mathcal{A}_t^{IB}$ and $\mathcal{A}_t^{IG}$ over all rounds $t \in [T]$, as described in Theorem 4.6. Using Lemma 4.7, we have that

$$\mathop{\mathbb{E}}_{\mathbf{r} \sim [0,1]^k, \mathcal{Q}} \left[ \sum_{t \in [T]} \Delta_{\mathcal{G}_t(\mathbf{r})}(\mu_t) \right] = \mathop{\mathbb{E}}_{\mathbf{r} \sim [0,1]^k, \mathcal{Q}} \left[ \sum_{t \in [T]} \sum_{i \in \mathcal{A}_t^{IG}} \Delta_{i, \sigma_t^{-1}(i)} \right]$$

$$= \mathop{\mathbb{E}}_{\mathbf{r} \sim [0,1]^k, \mathcal{Q}} \left[ \sum_{t \in [T]} \sum_{i \in \mathcal{A}_t^{IG}} \sum_{j \in \mathcal{A}} \Delta_{i,j} \mathcal{X}\left(\sigma_t(j) = i\right) \right]$$

$$\leq \mathop{\mathbb{E}}_{\mathbf{r} \sim [0,1]^k, \mathcal{Q}} \left[ \sum_{t \in [T]} \sum_{j \in \mathcal{A}} \sum_{i < j} \Delta_{i,j} \mathcal{X}\left(\sigma_t(j) = i\right) \right], \qquad (13)$$

where in the last inequality we restrict ourselves to arms $i < j$, where $\Delta_{i,j} > 0$.

Now using the results of Lemma 4.8, we can further upper bound (13) as

$$\mathop{\mathbb{E}}_{\mathbf{r} \sim [0,1]^k, \mathcal{Q}} \left[ \sum_{t \in [T]} \sum_{j \in \mathcal{A}} \sum_{i < j} \Delta_{i,j} \mathcal{X}\left(\sigma_t(j) = i\right) \right]$$

$$= \mathop{\mathbb{E}}_{\mathbf{r} \sim [0,1]^k, \mathcal{Q}} \left[ \sum_{t \in [T]} \sum_{j \in \mathcal{A}} \sum_{i < j} \Delta_{i,j} \mathcal{X}\left(\sigma_t(j) = i, T_j(t) \leq \ell_{i,j}\right) \right]$$

$$+ \mathop{\mathbb{E}}_{\mathbf{r} \sim [0,1]^k} \left[ \mathop{\mathbb{E}}_{\mathcal{Q}} \left[ \sum_{t \in [T]} \sum_{j \in \mathcal{A}} \sum_{i < j} \Delta_{i,j} \mathcal{X}\left(\sigma_t(j) = i, T_j(t) > \ell_{i,j}\right) \right] \right]$$

$$\leq \sum_{j > 1} \frac{16}{\Delta_{j-1,j}} \ln(T) + \frac{\pi^2}{3} \sum_{j > 1} \sum_{i=1}^{j-1} \Delta_{i,j}. \qquad (14)$$

By combining inequalities (12), (13) and (14), we can upper bound the regret as a function of the gaps as follows:

$$\alpha \text{OPT}(T) - \mathcal{R}^{IB}(T)$$

$$\leq \sum_{j>1} \frac{16}{\Delta_{j-1,j}} \ln(T) + \frac{\pi^2}{3} \sum_{j>1} \sum_{i=1}^{j-1} \Delta_{i,j} + \mathcal{O}(d_{\max} \cdot \text{rk}(\mathcal{M})) \quad \text{(gap-dependent regret)}.$$

In order to conclude the proof of the theorem, we would like to construct a regret bound that is independent of the gaps. The standard method is to partition the suboptimality gaps into "small" and "large" and, then, separately study their contribution to the regret. Specifically, for each $j \in \mathcal{A}$ and fixed $\epsilon > 0$, we define:

$$S_j = \{i < j \mid \Delta_{i,j} \leq \epsilon\} \text{ and } L_j = \{i < j \mid \Delta_{i,j} > \epsilon\}.$$

Starting again from (13) and noticing that the total regret due to small gaps can be at most $\epsilon \cdot T$ per arm, we have

$$\mathbb{E}_{\mathbf{r} \sim [0,1]^k, \mathcal{Q}} \left[ \sum_{t \in [T]} \sum_{j \in \mathcal{A}} \sum_{i<j} \Delta_{i,j} \mathcal{X}(\sigma_t(j) = i) \right]$$

$$= \mathbb{E}_{\mathbf{r} \sim [0,1]^k, \mathcal{Q}} \left[ \sum_{t \in [T]} \sum_{j \in \mathcal{A}} \sum_{i \in S_j} \Delta_{i,j} \mathcal{X}(\sigma_t(j) = i) \right] + \mathbb{E}_{\mathbf{r} \sim [0,1]^k, \mathcal{Q}} \left[ \sum_{t \in [T]} \sum_{j \in \mathcal{A}} \sum_{i \in L_j} \Delta_{i,j} \mathcal{X}(\sigma_t(j) = i) \right]$$

$$\leq \epsilon k T + \mathbb{E}_{\mathbf{r} \sim [0,1]^k, \mathcal{Q}} \left[ \sum_{t \in [T]} \sum_{j \in \mathcal{A}} \sum_{i \in L_j} \Delta_{i,j} \mathcal{X}(\sigma_t(j) = i) \right]. \tag{15}$$

We now focus only on the regret due to the large gaps, namely, the pairs $i, j$ such that $j \in \mathcal{A}$ and $i \in L_j$, which implies that $\Delta_{i,j} > \epsilon$. By exactly the same analysis as in the gap-dependent case, we can reach inequality (14), in the restricted case where the summations only include pairs of arms such that $\Delta_{i,j} > \epsilon$ (notice that we can apply Lemma 4.8 considering only the set $L_j$ of arms for each $j > 1$). In addition, using the fact that $\Delta_{i,j} \leq 1$ for any $i, j \in \mathcal{A}$, we have

$$\mathbb{E}_{\mathbf{r} \sim [0,1]^k, \mathcal{Q}} \left[ \sum_{t \in [T]} \sum_{j \in \mathcal{A}} \sum_{i \in L_j} \Delta_{i,j} \mathcal{X}(\sigma_t(j) = i) \right] \leq \sum_{j>1} \frac{16}{\epsilon} \ln(T) + \frac{\pi^2}{6} k(k-1). \tag{16}$$

By combining inequalities (15) and (16) with (12) and (13), we have

$$\alpha \text{OPT}(T) - \mathcal{R}^{IB}(T) \leq \epsilon k T + \frac{16k}{\epsilon} \ln(T) + \frac{\pi^2}{6} k(k-1) + \mathcal{O}(d_{\max} \cdot \text{rk}(\mathcal{M})).$$

Finally, by setting $\epsilon = 4\sqrt{\frac{\ln(T)}{T}}$, we get that

$$\alpha \text{OPT}(T) - \mathcal{R}^{IB}(T) \leq 8k\sqrt{T \ln(T)} + \frac{\pi^2}{6} k(k-1) + \mathcal{O}(d_{\max} \cdot \text{rk}(\mathcal{M})) \quad \text{(gap-independent regret)}.$$

Therefore, we can conclude that the expected reward collected by IB in $T$ rounds is at least

$$\left(1 - \frac{1}{e}\right) \text{OPT}(T) - \mathcal{O}\left(k\sqrt{T \ln(T)} + k^2 + d_{\max} \cdot \text{rk}(\mathcal{M})\right).$$

$\square$

# E    Tight examples for natural approaches

**Tight example for the naive greedy algorithm for MBB.**

**Lemma E.1.** *For any $d \geq 2$, there exists an instance of the full-information variant of the MBB problem (where the mean rewards are known a priori) such that the greedy strategy that plays a maximum mean reward independent set among the available arms collects a $\left(\frac{1}{2} + \frac{1}{2d}\right)$-fraction of the optimal expected reward.*

*Proof.* We consider an infinite time horizon and a graphic matroid based on the graph $G_d = (V_d, E_d)$, which is recursively defined as follows: Let $G_1 = (V_1, E_1)$ with $V_1 = \{u, v\}$, $E_1 = \{\{u, v\}\}$ and assume that the arm associated with edge $\{u, v\}$ has delay 1 and mean reward $1 - \epsilon$, for some $\epsilon > 0$. For the graph $G_d = (V_d, E_d)$, we have $V_d = V_{d-1} \cup \{u_d\}$ and $E_d = E_{d-1} \cup \{\{u, u_d\}, \forall u \in V_{d-1}\}$ (namely, $G_d$ is essentially the result of the join operation between $G_{d-1}$ and a single vertex graph). The arms that are associated with the edges of $E_d \setminus E_{d-1}$ all have delay equal to $d$ and mean reward equal to $1 - \frac{\epsilon}{d}$. The above recursive construction is illustrated in Figure 1.

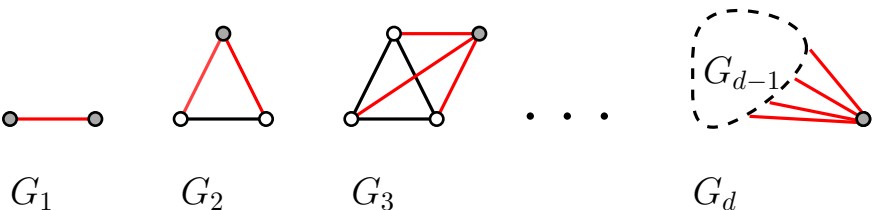

$$G_1 \qquad\qquad G_2 \qquad\qquad G_3 \qquad\qquad\qquad G_d$$

Figure 1: Recursive definition of $G_d$.

Consider now the arm-pulling schedule constructed by the greedy strategy. Let $T_p = E_p \setminus E_{p-1}$ be the new edges added at each step $p \in [d]$ in the recursive definition of $G_d$ (assuming that $E_0 = \emptyset$). Notice that for any integers $d \geq p_1 > p_2 \geq 1$ the edges of $T_{p_1}$ correspond to arms of higher mean reward than the edges of $T_{p_2}$. Therefore, the algorithm produces a periodic schedule of period $d$ as follows: Initially, the algorithm plays the $d$ arms of group $T_d$, collecting reward $d\left(1 - \frac{\epsilon}{d}\right) = d - \epsilon$. Notice that, by construction, these edges form a spanning tree in $G_d$ and, thus, no additional arm can be played at the same time step. In the second time step of the period, the arms of $T_d$ are blocked and the algorithm plays the arms of $T_{d-1}$ collecting $d - 1 - \epsilon$ reward. Again, this is the maximum reward independent set of $G_d$ among the available arms. The algorithm proceeds similarly in the following steps and collects an average reward of

$$\frac{\sum_{p=1}^{d}(p - \epsilon)}{d} = \frac{d \cdot (d+1)/2 - d\epsilon}{d} = \frac{d+1}{2} - \epsilon.$$

In the above example, the optimal arm-pulling sequence is to play at each time $t \in [T]$, one arm of each group $T_p$ for $p \in [d]$. Notice that by construction of the delays and at each time step, there always exists at least one arm per group that is available. Moreover, by definition of the graph $G_d$, any such selection of arms never contains a circuit and, thus, it is an independent set of the graphic matroid. The expected reward collected by the optimal algorithm at each step is $d - \epsilon \sum_{p \in [d]} \frac{1}{p} = d - \epsilon H(d)$, were $H(d) = \sum_{p \in [d]} \frac{1}{p}$.

In the above example, the ratio between the average reward collected by the greedy strategy and the optimal reward for $\epsilon \to 0$ becomes

$$\lim_{\epsilon \to 0} \frac{\frac{d+1}{2} - \epsilon}{d - \epsilon H(d)} = \frac{1}{2} + \frac{1}{2d}.$$

Therefore, by choosing large enough $d$, we can bring the approximation ratio of the above example arbitrarily close to $\frac{1}{2}$. $\qquad\square$

**Tight example for the naive greedy algorithm for RSW.**

**Remark E.2.** *The greedy approach of choosing $\mathcal{A}_t$ to be the set of all available elements at round $t \in [T]$ can be as bad as a $\frac{1}{k}$-approximation. In order to see that, consider the monotone (budget-additive) submodular function $f(S) = \min\{|S|, 1\}$. Let $k$ be the number of elements with delay $d_i = k$ for each $i \in \mathcal{A}$. Assuming an infinite time horizon, the optimal strategy collects an average reward of $1$, simply by choosing one element at a time in a round-robin manner. However, the average reward of the greedy approach in this case is $\frac{1}{k}$.*

**Tight example for independent sampling for RSW.**

**Remark E.3.** *The independent randomized sampling approach of adding each arm $i$ to $\mathcal{A}_t$ independently with probability $\frac{1}{d_i}$, if available, can be as bad as a $(1 - \frac{1}{\sqrt{e}})$-approximation. Consider the*

*same setting as in Remark E.2, where for $T \to \infty$ the optimal average reward is $1$. However, the average expected reward of the independent randomized sampling strategy is $1 - (1-p)^k$, where $p = \frac{1}{2k-1}$ is the probability that each element is selected at each round (in stationarity). For $k \to \infty$, we have that $1 - (1-p)^k \to 1 - e^{-\frac{1}{2}} \approx 0.393$.*