# OpenReview forum: "Recurrent Submodular Welfare and Matroid Blocking Semi-Bandits"
_NeurIPS.cc/2021/Conference — NeurIPS 2021 Poster_

### Official Review · Reviewer_GPpn · 2021-07-17

**Rating:** 8
**Confidence:** 4

**Summary:**

This paper focuses on matroid blocking semi-bandit and connected this problem to the problem of recurrent submodular welfare maximization.

**Limitations And Societal Impact:**

The authors discussed the limitations and potential negative social impact of their work.



**Main Review:**

1) The author tried to motivate the subject from a theoretical and practical point of view, but they didn't add any simulation to show the performance of their method from a practical point of view.
2) In the introduction, the author reviewed the previous works. Still, related papers like [1] have not been appropriately discussed.
3)the author successfully addressed the issue and recommended the algorithm.
4) The paper is well-organized and well-written.
5) The proof is mathematically correct.
[1]Chen, Lin, Andreas Krause, and Amin Karbasi. "Interactive Submodular Bandit." Advances in Neural Information Processing Systems 30 (2017).

----------- Post Rebuttal-------------

I have read all the comments, and I don't change my score.

**Time Spent Reviewing:**

6

---

> ### Author Response · Authors · 2021-08-09
> **Answer to Reviewer GPpn**
>
> We would like to thank the reviewer for the valuable feedback. We now address the main comments/questions of the review:
>
> **Missing reference.** Thank you very much for pointing out the missing reference. This work is indeed related to ours. We will cite/discuss the paper (and this line of work in general) in the revision.
>
> **Empirical evaluation.** In our paper, we focus on the theoretical side of the problem (both in terms of approximation and of regret analysis). We believe that our technique of interleaved rounding can be further applied to multi-armed bandits with different reward functions (other than blocking), where the idea of separating scheduling and learning can facilitate the regret analysis. An experimental evaluation and comparison of our algorithm with other heuristics (mentioned in the paper) could be an interesting future direction.

---

### Official Review · Reviewer_b67u · 2021-07-18

**Rating:** 6
**Confidence:** 3

**Summary:**

The paper studies an extension of the matroid bandit problem where each arm $i$ is blocked for a known $d_i$ rounds after it is played, and this is referred to as the matroid blocking semi-bandit (MBB) problem. When the arm distributions are known, the authors reduce the optimization problem to a recurrent submodular welfare (RSW) problem. The authors first propose a 1-1/e approximation to the RSW problem, then show that for MBB with unknown reward distributions, using the upper confidence bounds (UCB) instead of the known mean values would give a sublinear approximation regret with approximation ratio of 1-1/e. The main contribution of the paper is to achieve 1-1/e approximation, while an easy adaptation from some other work only gives 1/2 approximation.

**Ethical Concerns:**

There is no ethical concern on this work.

**Limitations And Societal Impact:**

The research is a theory work, and there is no perceived negative societal impact.

**Main Review:**

The main contribution of the paper is to combine matroid bandit and blocking bandit to propose a solution to the matroid blocking semi-bandit problem, and it achieves 1-1/e approximation regret rather than the easier 1/2-approximation regret that could be obtained by some easier extension to some other work. Along the way, the authors define the recurrent submodular welfare problem and provide a 1-1/e approximation solution, and this could be of independent interest.

Technically, the algorithms proposed for RSW and MBB are based on some scheduling scheme, a technique used in the past studies. The online learning version uses the standard UCB idea. Overall, the paper presents an interesting problem with a nice solution. The intermediate problem of RSW is also interesting. The presentation is clean mostly. However, it is a bit unclear how much technical novelty is in the paper, since it seems that the results mainly come from several algorithmic ideas in the past.

More detailed comments:

- Line 47, "collects their sum". It is better explicitly say that "collects their sum as the reward for this round". Otherwise, it is unclear what is exactly the reward for each round.

- Line 101, what is $x_i$?

- Line 195, Theorem 1.4. It would be better if the authors also include the gap-dependent bound in the theorem.

- Line 212-213. The formula $f^+(x)$ has no explanation. Notations such as $1_S$ is not explained. What is the intuitive meaning of this function?

- Line 263-264, "For integral solutions, the above objective is equivalent to ...." Why is this so?

- Line 297, notation $\mathcal{X}$ is not explained.

- Line 298-299. Normally in online learning, "full-information" would mean that in every round the agent would receive feedback for all arms, but here "full-information" means that the arm distributions are known a priori. I believe the latter case is better called the offline setting, vs. the online learning setting.

- Line 336, using notation $\mathcal{R}$ to denote the randomness of the reward is confusing with the same notation used to denote the expected reward.

- Line 339. It is unclear to me why w.l.o.g we only focus on the case where the sequences of sampled arms are identical. What if they are not identical?

- Line 392 - 399. This paragraph needs some repeated reading to get the right idea. I guess what the authors mean is that MBB can be reduced to RSW, but the reverse may not be true, so MBB could have better approximation ratio, like having a PTAS. Also, throughout the paper, the authors often refer to MBB as an offline problem and talks about its approximation algorithm, while MBB is defined originally as an online learning algorithm. I guess sometimes the authors refer MBB with the known reward distributions. I suggest the the authors make the explicit distinction between the offline problem and the online problem, such as using MBB-offline to refer to the offline (full-information in their term) setting.

========= Post rebuttal comments ===============

I have read the authors' feedback and other review comments. As indicated in my review, my main concern is the technical novelty of the paper. The authors' response tried to address this, but it is still not entirely clear to me. However, in general, I think the paper has merit, and would like to encourage the authors to revise and further improve the paper, whether or not it is accepted at this conference.

**Time Spent Reviewing:**

4 hours

---

> ### Author Response · Authors · 2021-08-09
> **Answer to Reviewer b67u**
>
>
> We would like to thank the reviewer for the valuable feedback. We now address the main comments/questions of the review:
>
> **Technical novelty.** Regarding the technical novelty of our work, we need to emphasize that, to the best of our knowledge, *our algorithmic technique has not been used in past studies*. As we discuss in our paper (lines 157-161), our method resembles the interleaved rounding technique [25] only in spirit (our algorithm is not even rounding-based). Indeed, our problem poses important additional challenges including the matroid constraints (as opposed to [25] where at most one arm can be played at each round) and the fact that the hard blocking constraints cannot tolerate any variance in the produced schedule (as opposed to the increasing concave reward functions of [25]). The two methods are very different implementations of the idea of time-correlated arm sampling. As far as the learning part of our work is concerned, our technique allows the decoupling of the scheduling part from the learning part, which in turn allows the use of more standard UCB arguments for bounding the regret.
>
>
> **Answers to detailed comments.**
>
> - We will change the phrase "collects their sum" into "collects their sum as the reward for this round".
> - $\frac{1}{x_i}$ is the "optimal" playing frequency for arm $i$ computed via a mathematical formulation in [25]. $x_i$ indicates that in an "optimal" solution, arm $i$ must be played every $x_i$ rounds.
> - The reason for not including the gap-dependent regret bound in Theorem 1.4, is that we do not want to introduce additional notation in the contribution section of our paper. We will add this, if space permits.
> - Due to space constraints, we were not able to discuss the formula of $f^+(x)$. However, the concave closure is a standard notion in the analysis of submodular functions. Intuitively, for any point $x$, $f^+(x)$ is the maximum of $\sum_{S \subseteq \mathcal{A}} a_Sf(S)$, over any distribution $\alpha$ on the subsets of elements, such that the marginal probability of any element $i$ being in the sampled set is exactly $x_i$. Further, $1_S$ is a vector in {$\{0,1\}$}$^{|\mathcal{A}|}$, having $1$ for each coordinate $i \in S$, and $0$, otherwise. We will add a remark for the above definitions, if space allows.
> - Notice that the multi-linear extension, $F(x)$, and the concave closure, $f^+(x)$, have exactly the same value when $x$ is an integral vector.
> - Thank you for catching that the notation $\mathcal{X}$ is not defined. $\mathcal{X}(E)$ is the indicator function that becomes $1$ if the event $E$ holds, and $0$, otherwise. We will add a definition in the revision.
> - You are correct that the term "full-information" can be a bit misleading. However, even when the player knows the expected rewards, the absence of knowledge of the time-horizon makes it hard for us to call this case "offline". We will add a remark clarifying the term "full-information" and distinguishing it from its usual definition.
> - You are correct that the notation $\mathcal{R}$ to denote the randomness due to the reward realizations can be confusing. We will change it.
> - The reason why we can focus on the case where the sequences of sampled arms are identical is a very important feature of our algorithm. The reason we are able to do this is that the sequences of sampled arms in the full-information and the bandit case of our problem are identically distributed. Indeed, for each arm, the sequence depends on its delay and the outcome of a Uniform[0,1] random variable, while it is decided before the beginning of the online phase (i.e., independently of the trajectory of observed rewards). Thus, in order to upper bound the regret (i.e., the difference between the expected reward of the full-information and the bandit algorithm), it suffices to upper bound the regret "point-wise", for any sequence of sampled arms (see also the proof of Lemma 4.4, p. 19 of the Supplementary Material).
> - You are correct that we refer to MBB to denote both its full-information and its bandit version, depending on the context. Thank you for catching this confusion. We will make sure to clarify each reference in the revision.

---

### Official Review · Reviewer_hwtn · 2021-07-20

**Rating:** 7
**Confidence:** 4

**Summary:**

The authors consider the problem of Matroid Blocking Semi-Bandits. In this setting, there are $k$ arms, an unknown time horizon $T$ and a matroid dictating which sets of arms can be pulled at each step. Pulling each arm yields a reward, which comes from an unknown probability distribution with mean $\mu_i$. Moreover, each arm $i$ remains blocked after use for $d_i$ steps. Observing only the rewards from the used arms, the goal is to maximise the sum of all rewards collected over the time horizon $T$.

There are 3 main results, which can be better explained by a minor detour in the full information setting (where the $\mu_i$ are known). In this setting, the first approach one would follow is to ignore the $d_i$’s and just chose the independent set maximising the current rewards, among the arms that are not blocked. However, this is not optimal (for general matroids): the catch is that the underlying matroid indirectly affects the rewards of each arm, by restricting which other arms can be used together. To get around this, the authors establish a reduction between this setting and what they call the Recurrent Submodular Welfare (RSW), which entails maximising a monotone submodular function over $T$ steps, but where each element has a similar delay $d_i$ before it can be selected again.

The RSW problem can be solved by a clever randomisation: rather than choosing each element with probability $1/d_i$, the authors interleave the elements by adding a random delay $r_i \in [0,1]$ and selecting the item whenever the interval [t/d_i + r_i, (t+1)/d_i + r_i] contains an integer. This simple step takes care of most of the inefficiency and the unknown time horizon $T$ is also handled: essentially, just the last step might be suboptimal and this effect decreases with $T$. This algorithm is optimal for this setting, with a $1-1/e$ approximation.

The full information setting can be handled by selecting arms a subset of arms that are available in the RSW case, but doing it greedily and stopping once an independent set is found, to satisfy that hard matroid constraints. The reduction is completed by using the weighted rank function, which converts an additive function with matroid constraints into a monotone submodular one. As such, this algorithm is also optimal with an approximation of $1-1/e$ (and some minor terms relating to $\max d_i$).

The final step into the bandit setting can be performed because the learning of the $\mu_i$ and the random selection of elements are decoupled: the RSW interleaving technique is ‘static’ and does not update its schedule as the algorithm runs. Therefore, a modification of UCB can be used to balance learning the $\mu_i$’s with exploiting them, while observing the schedule used by RSW. This also yields an optimal (1-1/e) approximation (with an additional  $O(k\sqrt{T \log T} + k^2  + \max d_i \cdot r)$ additive loss, where $r$ is the matroid rank).

**Limitations And Societal Impact:**

The authors have adequately addressed the limitations and social impact.

**Main Review:**

The paper is very well written and provided an clear description of related work, in particular comparing with Blocking Bandits and Kleinberg and Immorlica [25], which also use a different interleaved rounding technique for a related setting where each arms ‘improves’ as it remains unused. Moreover, the authors have done a great job condensing the paper into 9 pages, with most of the intuition left intact.

The paper is technically strong, with the designed algorithms looking simple but containing subtleties that can be easily missed: there are many similar algorithms that almost work but are not optimal, which is what the authors claim as the main contribution of the paper. In terms of techniques, the authors have an interesting use of Convex Programming and the concave closure of a submodular function, both of which have received significant attention by the sumbodular community in recent years (although using the multilinear extension and rounding is more common).  The only criticism could be about the significance of the setting: why is it important to bridge the gap from $1-1/e^{0.5}$ (with a very simple algorithm) to $1-1/e$?

There are no experiments, which is could be ok for this type of paper, given that finding real data sets with matroid constraints and bandit feedback might be difficult. However, it would be nice to have some evidence (even on synthetic data) that the obvious algorithm is not as good in practice as the one designed here.

**Time Spent Reviewing:**

5

---

> ### Author Response · Authors · 2021-08-09
> **Answer to Reviewer hwtn**
>
> We would like to thank the reviewer for the valuable feedback. We now address the main comments/questions of the review:
>
> **Bridging the gap between $1- \frac{1}{\sqrt{e}}$ and $1 - \frac{1}{e}$.** Notice that the simple algorithm based on independent sampling can only guarantee that we collect (in expectation) a $0.393$-fraction of the optimal expected reward. On the other hand, our algorithm (which is also simple and computationally equally demanding) guarantees a $0.632$-fraction of the optimal expected reward. The guarantee of our algorithm is significantly higher, which is important for applications where maximizing the total expected reward is crucial. Besides, for instances running over a large time horizon, it is important to have a bandit algorithm that converges to (i.e., learns) the best possible full-information strategy.
>
> **Empirical evaluation.** In our paper, we focus on the theoretical side of the problem (both in terms of approximation and of regret analysis). We believe that our technique of interleaved rounding can be further applied to multi-armed bandits with different reward functions (other than blocking), where the idea of separating scheduling and learning can facilitate the regret analysis. An experimental evaluation and comparison of our algorithm with other heuristics could be an interesting future direction.

---

### Official Review · Reviewer_WQ2T · 2021-07-20

**Rating:** 6
**Confidence:** 3

**Summary:**

This paper introduces matroid blocking semi-bandits (MBB), which generalizes blocking bandits and matroid bandits. To solve the full-information variant of MBB, the authors reduce it to Recurrent Submodular Welfare (RSW) and propose a novel technique of interleaved scheduling to achieve a (1-1/e)-approximation. For the bandit setting of MBB where the mean arm rewards are unknown, based on the result from the full-information setting, they propose an UCB-type algorithm with regret guarantees.

**Limitations And Societal Impact:**

1) Though the authors discuss the main steps of reducing MBB to RSW on line 127-150, more details on why player A's task is a special case of RSW are missing (line 298-308 only discuss the algorithm and the theorem).
2) Once the full-information setting is solved, the UCB-based algorithm for the bandit setting is somewhat standard. Though the regret analysis becomes harder and the authors use some properties of matroids, the remaining parts are based on [25].

**Main Review:**

1) Originality:
One key idea of this paper is reducing the full-information MBB to RSW, which is novel and serves as the cornerstone for the bandit setting. To solve RSW, the authors develop the interleaved scheduling technique motivated by [25] to design a (1-1/e)-approximation algorithm. They also prove the (1-1/e)-approximation ratio is asymptotically tight. Based on these results, they propose a UCB-based algorithm for the bandit setting. Their regret analysis uses the strong basis exchange property of matroids to decompose regret, then follows [28] to derive the regret guarantee.

2) Quality:
This paper is technically sound. Most of the claims are supported by theoretical analysis.

3) Clarity:
This paper is clearly written and well organized.

4) Significance:
The main technical contributions of this paper are: (1) Reducing the full-information MBB to RSW; (2) (1-1/e)-approximation algorithm for RSW; (3) Using the strong basis exchange property of matroids to decompose regret for the regret analysis.


**Time Spent Reviewing:**

3

---

> ### Author Response · Authors · 2021-08-09
> **Answer to Reviewer WQ2T**
>
> We would like to thank the reviewer for the valuable feedback. We now address the main comments/questions of the review:
>
> **Player A's task is a special case of RSW.**
> The reason why Player A's task is a special case of RSW is due to the fact that, at each round, Player B's optimal strategy is to play the maximum expected reward independent set among the arms chosen by Player A, which is a monotone submodular function. This is stated in lines 304-306 and is described in more details in the contribution section (lines 141-150). We will add more details above Theorem 4.2, if space permits.
>
> **Regret analysis.** As we discuss in the paper, our main contribution to the regret analysis is the technique of interleaved scheduling. This technique not only provides an improved approximation guarantee (which, as we show, is optimal for the RSW problem), but also provides an elegant way to analyze the regret. Indeed, the reason we are able to apply more standard UCB arguments (e.g., as in [28]) is that, as product of our algorithmic technique, we are able to decouple the scheduling from the learning part. Without our technique, it is unclear to us how to control the $(1-1/e)$-regret.

---

### Official Review · Reviewer_F9ZW · 2021-07-28

**Rating:** 6
**Confidence:** 3

**Summary:**

This paper considers the problem of matroid semi-bandits with an additional condition that each arm becomes unavailable for a fixed duration after it is played. The paper first considers an "offline" version of the problem where the reward for each arm is already known and the goal is to design an optimal schedule for playing the arms. A randomized greedy algorithm is proposed which is (1-1/e)-competitive to an optimal algorithm. The paper then shows that this algorithm can be extended to the `online' version of the problem when the rewards are unknown. This is because the set of arms played in this greedy schedule only depends on the delays and does not depend on the rewards. This online algorithm suffers an additional regret of O(K\sqrt{T} + K^2) in comparison to the best offline solution.

**Limitations And Societal Impact:**

The broader impact section is not provided. I do not see any immediate negative impacts.

**Main Review:**

Sub-optimality of Regret: The regret upper bound presented in the paper seems to be weak along a few dimensions. Firstly, there is a K^2 dependence in the regret bound which is large for any practical application. Is this K^2 dependence really needed for this problem or is it just an artefact of the algorithm/analysis? I would like to believe that it is the latter since \Delta_{ij} contains information about \Delta_{kj} which can perhaps be exploited. Secondly, there is a dependence on d_{\max} which can be large if a few arms are blocked for a long time.

Technical Depth: Even though the theoretical results are sound, I did not find much technical depth in the algorithm and analysis. The analysis of (1-1/e) competitiveness in the "offline" case follows the usual multilinear extension/concave closure "correlation gap" technique from submodular optimization. In the "online" extension I liked the use of "strong basis exchange" property of matroids, however, the rest of the argument follows from an "UCB-type" analysis of the gaps/regret of the arms.

Known delays: The assumption that the delays for each arm is known is not realistic. Is it possible to extend this algorithm to the case when delays are stochastic in nature and only the expected delay for each arm is known?

Summary of review: This paper proposes an interesting problem and proposes sound theoretical results for this problem. However, there are several issues outlined above due to which I think this paper is borderline.





**Time Spent Reviewing:**

7

---

> ### Author Response · Authors · 2021-08-09
> **Answer to Reviewer F9ZW**
>
> We would like to thank the reviewer for the valuable feedback. We now address the main comments/questions of the review:
>
> **$k^2$ dependence in regret.** First, notice that in the case of standard (non-blocking) matroid bandits, the analogous additive dependence is $k \cdot r$, where $r$ is the rank of the matroid. Thus, for $r = O(k)$, our dependence is the same as in standard matroid bandits. Further, observe that the $k^2$ dependence is not multiplied with the time horizon $T$ and, thus, for the practical scenario where $T \log (T) > k^2 $, it becomes negligible (up to constant factors). In any other case, exploring the necessity of this dependence in the regret bound is an interesting future direction. On the one hand, indeed your argument that $\Delta_{ij}$ carries information about $\Delta_{kj}$ is reasonable. On the other hand, notice that, as opposed to standard matroid bandits where the maximum independent set (of $r$ arms) is fixed throughout the time horizon, in the blocking setting the complete order of mean rewards is needed in order to play ``optimally'' (i.e., as the full-information algorithm), given that the set of sampled arms changes dynamically over time.
>
> **$d_{\max}$ dependence in regret.** Notice that the additive $d_{\max}$ dependence in the regret guarantee becomes negligible (up to constants) for the practical scenario where $d_{\max}$ as large as $\mathcal{O}(\frac{k}{r} \sqrt{T \log T})$. This dependence has nothing to do with the learning part of our work, but originates from the use of the convex program as an upper bound (see the additive loss in Lemma 3.3). Indeed, the convex program we use involves as variables the average expected number of times any arm is played and, thus, is unable to capture limiting packing issues that can result from a very small time horizon. In order to avoid this dependence, one could design a convex-programming based randomized rounding algorithm, using a convex program that includes an assignment variable for each pair (arm, round). However, this approach is not only computationally inefficient, but also requires knowledge of the time horizon. Finally, it is unclear to us how this approach can be adapted to the learning setting without significantly complicating the bandit algorithm.
>
>
> **Extension to stochastic delays.** As we remark in the conclusion of our paper, our algorithm can be easily adapted to provide an $\mathcal{O}(1)$-approximation for the case of stochastic delays. We briefly describe a simple adaptation that only requires knowledge of the expected delays: Let ${E}[d_i]$ be the expected delay of arm $i$. It can be proved that our convex program remains an (approximate) upper-bound on the optimal expected reward, if we replace the arm (fixed) delays with the expected delays. We adapt our interleaved rounding technique in a way such that each arm is sampled every (roughly) $2 \cdot E[d_i]$ rounds. At each round, any sampled arm is played only if it is available. Notice that, by Markov inequality, since we sample the arm every $2 \cdot E[d_i]$ rounds, every time it is sampled, it is available with probability at least half. The above simple adaptation provides a $\frac{1}{4}\left(1 - \frac{1}{e}\right)$-approximation for the problem, using only knowledge of the expected delays. Further, improved constants can be achieved by balancing the sampling rate and availability using knowledge of the whole distribution (or additional moments). Improving the approximation guarantee for this case is an interesting open question.

---

### Decision · Program_Chairs · 2021-09-27

**Decision:**

Accept (Poster)

**Comment:**

This paper on bandits merged together many different aspects, submodularity on matroid, blocking bandits...

We are not necessarily excited by papers that combine together exciting concepts, but this specific papers had other merits as the base offline problem is also not that trivial and requires different (new) techniques.

Overall, the reviewers were all positive, the paper is well written, clear and easy to read. It can therefore be accepted at NeurIPS